# Disrupting TSLP–TSLP receptor interactions via putative small molecule inhibitors yields a novel and efficient treatment option for atopic diseases

Partho Protim Adhikary [1], Temilolu Idowu [1], Zheng Tan[1], Christopher Hoang[1], Selina Shanta[1], Malti Dumbani[2], Leah Mappalakayil [1], Bhuwan Awasthi [1], Marcel Bermudez [2,3], January Weiner[4], Dieter Beule[4], Gerhard Wolber[2], Brent DG Page [1✉] & Sarah Hedtrich [1,4,5,6✉]

## Abstract

**Thymic stromal lymphopoietin (TSLP) is a key player in atopic diseases, which has sparked great interest in therapeutically targeting TSLP. Yet, no small-molecule TSLP inhibitors exist due to the challenges of disrupting the protein–protein interaction between TSLP and its receptor. Here, we report the development of small-molecule TSLP receptor inhibitors using virtual screening and docking of >1,000,000 compounds followed by iterative chemical synthesis. BP79 emerged as our lead compound that effectively abrogates TSLP-triggered cytokines at low micromolar concentrations. For in-depth analysis, we developed a human atopic disease drug discovery platform using multi-organ chips. Here, topical application of BP79 onto atopic skin models that were co-cultivated with lung models and Th2 cells effectively suppressed immune cell infiltration and IL-13, IL-4, TSLP, and periostin secretion, while upregulating skin barrier proteins. RNA-Seq analysis corroborate these findings and indicate protective downstream effects on the lungs. To the best of our knowledge, this represents the first report of a potent putative small molecule TSLPR inhibitor which has the potential to expand the therapeutic and preventive options in atopic diseases.**

**Keywords** TSLP; Atopic Diseases; Atopic Dermatitis; Organ-on-chip; Small Molecule Inhibitor
**Subject Categories** Immunology; Pharmacology & Drug Discovery; Skin

## Introduction

Epithelium-derived thymic stromal lymphopoietin (TSLP) is a master regulator of Th2-driven inflammation. As such, it is distinctly increased in skin and lung tissue of atopic dermatitis (AD) patients and asthmatics, respectively (Peng and Novak, 2015; Werfel et al, 2016; Roan et al, 2019). AD is the most common inflammatory skin disease worldwide with currently up to 25% of children and 1–3% of adults being affected (Weidinger et al, 2018). Besides genetic predispositions, environmental and immunological factors contribute to AD manifestation and pathogenesis rendering it a multifactorial disease. Defects in the epidermal barrier of human skin favor sensitization and strong inflammatory responses to external stimuli, ultimately triggering a systemic response (Bieber, 2022). These skin barrier defects may be inherited. Prominent examples are mutations in the filaggrin gene, which have been identified in 10–50% of AD patients and are a major predisposing factor for AD (Palmer et al, 2006).

Notably, ≤50% of AD patients with moderate-to-severe phenotype undergo the atopic march meaning concomitant sensitization to indoor and later outdoor allergens, progressing to asthma and allergic rhinitis at later time points (Boguniewicz and Leung, 2011; Leyva-Castillo et al, 2013). Although the underlying mechanism remains largely unknown, the contribution of TSLP has been firmly established (Demehri et al, 2009; Zhang et al, 2009; Han et al, 2012; Leyva-Castillo et al, 2013; Han et al, 2017).

TSLP is an IL-7-like cytokine that exerts its biological activities by binding to its receptor. The TSLP–TSLPR heterodimer then recruits the IL-7 receptor α for active receptor complex formation (Zhong et al, 2014; Verstraete et al, 2017). This receptor complex is expressed by a wide range of immune cells including dendritic cells (DCs), macrophages, and T cells but also epithelial cells and neurons (Zhong et al, 2014). TSLP binding activates the TSLPR complex, which in turn activates protein tyrosine kinases and transcription factors including STAT3, STAT5, STAT6, and GATA3 which promote the expression of inflammatory cytokines such as IL-4 and IL-13 (Arima et al, 2010; Rochman et al, 2018). Moreover, TSLP activates DCs that subsequently prime human CD4[+] T cells into Th2 cytokine-producing cells in local lymph nodes (Soumelis et al, 2002; Watanabe et al, 2004; Ebner et al, 2007). TSLP signaling in CD4[+] T cells is also required for

[1]Faculty of Pharmaceutical Sciences, The University of British Columbia, Vancouver, BC, Canada. [2]Institute of Pharmacy, Freie Universität of Berlin, Berlin, Germany. [3]Institute of Pharmaceutical and Medicinal Chemistry, Westfälische Wilhelms-Universität Münster, Münster, Germany. [4]Berlin Institute of Health at Charité - Universitätsmedizin Berlin, Germany Charité - Universitätsmedizin Berlin, Berlin, Germany. [5]Department of Infectious Diseases and Respiratory Medicine, Charité—Universitätsmedizin Berlin, corporate member of Freie Universität Berlin and Humboldt Universität zu Berlin, Berlin, Germany. [6]Max-Delbrück Center for Molecular Medicine in the Helmholtz Association (MDC), 13125 Berlin, Germany. ✉E-mail: brent.page@ubc.ca; sarah.hedtrich@bih-charite.de

memory formation after Th2 sensitization (Wang et al, 2015) and the activation of group 2 innate lymphoid cells, which further drive the pathogenesis of inflammatory skin diseases (Kim et al, 2013). Recent data also indicate that TSLP triggers transcriptional changes in Th2 cells skewing them towards pathogenic phenotypes that produce greater amounts of pro-inflammatory cytokines than conventional Th2 cells (Rochman et al, 2018).

Two isoforms of human TSLP have been identified: short and long TSLP. The short isoform (sfTSLP; 60 - 63 amino acids) is constitutively expressed, critical for epithelial homeostasis, and does not bind to the TSLPR. In contrast, lfTSLP (long isoform, 159 amino acids) is inducible, can be massively upregulated in atopic diseases, and promotes inflammation rendering both, lfTSLP and TSLPR, putative therapeutic targets (Fornasa et al, 2015; Adhikary et al, 2021). Accordingly, the inhibition or knockout of TSLP signaling has shown great therapeutic potential (Cianferoni and Spergel, 2014).

The knowledge about the central role of TSLP in atopic diseases has sparked interest in therapeutically targeting TSLP. In fact, the anti-TSLP antibody tezepelumab is now being approved for the treatment of severe asthma (Menzies-Gow et al, 2023).

However, small-molecule TSLP inhibitors, which would allow a topical, non-invasive application, have not yet been developed as disrupting protein–protein interactions with small molecules is challenging. Nonetheless, in addition to lower costs of small molecules versus biologics, a topical application appears favorable since TSLP drives the initial sensitization for allergic asthma locally in the skin (Leyva-Castillo et al, 2013; Rochman et al, 2018; Segaud et al, 2022).

Hence, here we report the development of a topically applicable, putative small-molecule inhibitor of human TSLPR via a structure-based virtual screen and chemistry-based optimization, for the treatment of AD and, potentially, the prevention of the atopic march. Following initial screening and docking studies of 1,524,680 compounds, twelve virtual top hits were selected for initial in vitro validation. Based on those screening results, ligand-based virtual screening was performed using the NCI-DTP (National Cancer Institute - Developmental Therapeutics Program, USA). Eighty-four compounds were identified and subjected to in vitro screening. Building onto those results, novel inhibitors were synthesized which were then tested for efficacy and safety in primary human skin, dendritic cells as well as CD4 + T cells. BP79 emerged as the lead compound and was extensively characterized for its TSLPR inhibitory activity via western blots, proximity ligation assay, in vitro thermal shift assay, and kinome screening.

For further testing, we developed and leveraged a human-based atopic diseases drug discovery platform by combining bioengineered atopic-like skin disease models and 3D bronchial epithelial models on a microfluidic multi-organ chip. This setup enabled us to study the preclinical efficacy and safety of our lead compound in a complex human-like environment. This has great translational potential due to the avoidance of species-related differences which are prominent and challenging in the TSLP context.

To the best of our knowledge, this represents the first report of a potent and safe small-molecule TSLPR inhibitor which provides the opportunity not only to expand the therapeutic and preventive options in atopic diseases, but could also serve as a starting point for further developments in targeting TSLP as a central player of inflammation.

# Results

## Identification of hit compounds from chemical libraries

The ternary signaling complex of TSLP with its receptor forms in two steps: First, TSLP interacts with TSLPR to form a binary complex through distinct interactions and electrostatic complementarity. TSLP displays a positively charged surface and interacts with the negatively charged interdomain elbow region of TSLPR to form *site I*. Formation of *site I* is essential for the second step, in which the TSLP:TSLPR complex recruits IL-7Rα, thus, forming the signaling complex (Verstraete et al, 2014; Verstraete et al, 2017). Hereby, TSLP establishes a T-shaped ternary complex through two extensive interaction interfaces, namely *site I* (TSLP:TSLPR) and *site II* (TSLP:IL-7Rα). TSLP bridges the two receptors and *site III* (TSLPR:IL-7Ra) is formed involving the interaction between the membrane proximal residues of the two receptors (Verstraete et al, 2014; Verstraete et al, 2017). We therefore exploited structural information from the crystal structure of the ternary complex (PDB (Berman et al, 2000) code:5J11 (Verstraete et al, 2017)) and developed a 3D pharmacophore (Wolber and Langer, 2005) for virtual screening based on the interaction of TSLP with TSLPR (*site I*) aiming to identify small molecules that hinder TSLPR complex formation (Fig. 1A,B).

Pocket detection was applied to identify druggable binding sites on the surface of the complex. Cavity analysis indicated that the TSLP:TSLPR interface contains more druggable pockets than *site II* and *site III*. A thorough analysis of the crystal structure revealed key electrostatic complimentary interactions between Arg150, Arg153 of TSLP with Asp92 of TSLPR (Verstraete et al, 2014; Verstraete et al, 2017). Available mutational and fragment-based inhibitor identification studies also reported the integral role of these key residues, thus, solidifying our rationale to proceed with *site I* (Verstraete et al, 2014; Van Rompaey et al, 2017; Verstraete et al, 2017). Careful consideration of all information from cavity detection, mutation data, and docking studies enabled us to develop 3D pharmacophore models using LigandScout 4.4 (Wolber et al, 2006; Wolber and Langer, 2005) that specifically represent binding hotspots at the TSLP:TSLPR interface (Fig. 1A). Two 3D pharmacophores, one representing ligand interaction patterns for TSLPR and the other targeting TSLP, were developed and used for virtual screening to identify potential small-molecule binders inhibiting TSLP:TSLPR complex formation.

A library of 1,524,680 purchasable compounds from Specs (Delft, Netherlands) and VitasM (Hongkong, China) was screened using these two 3D pharmacophores. A total of 3838 and 56,309 virtual hits were obtained (Fig. 1B). Virtual hits were filtered by molecular weight (>250 Da), rotatable bonds (<7), and a minimum of 5 matching chemical features from the 3D pharmacophore model for further enrichment. The filtering resulted in the remaining 2864 and 1196 molecules as potential inhibitors for TSLPR and TSLP, respectively. To assess whether virtual hits fulfill the previously defined interaction patterns represented by the 3D pharmacophore, molecular docking experiments were performed by docking them into the prepared protein structure (PDB (Berman et al, 2000) code: 5J11 (Verstraete et al, 2017)) using GOLD v5.2 (Jones et al, 1997). The resulting poses were compared with the 3D pharmacophore alignment from the virtual screening, and similar conformations were prioritized. Subsequently, the selected

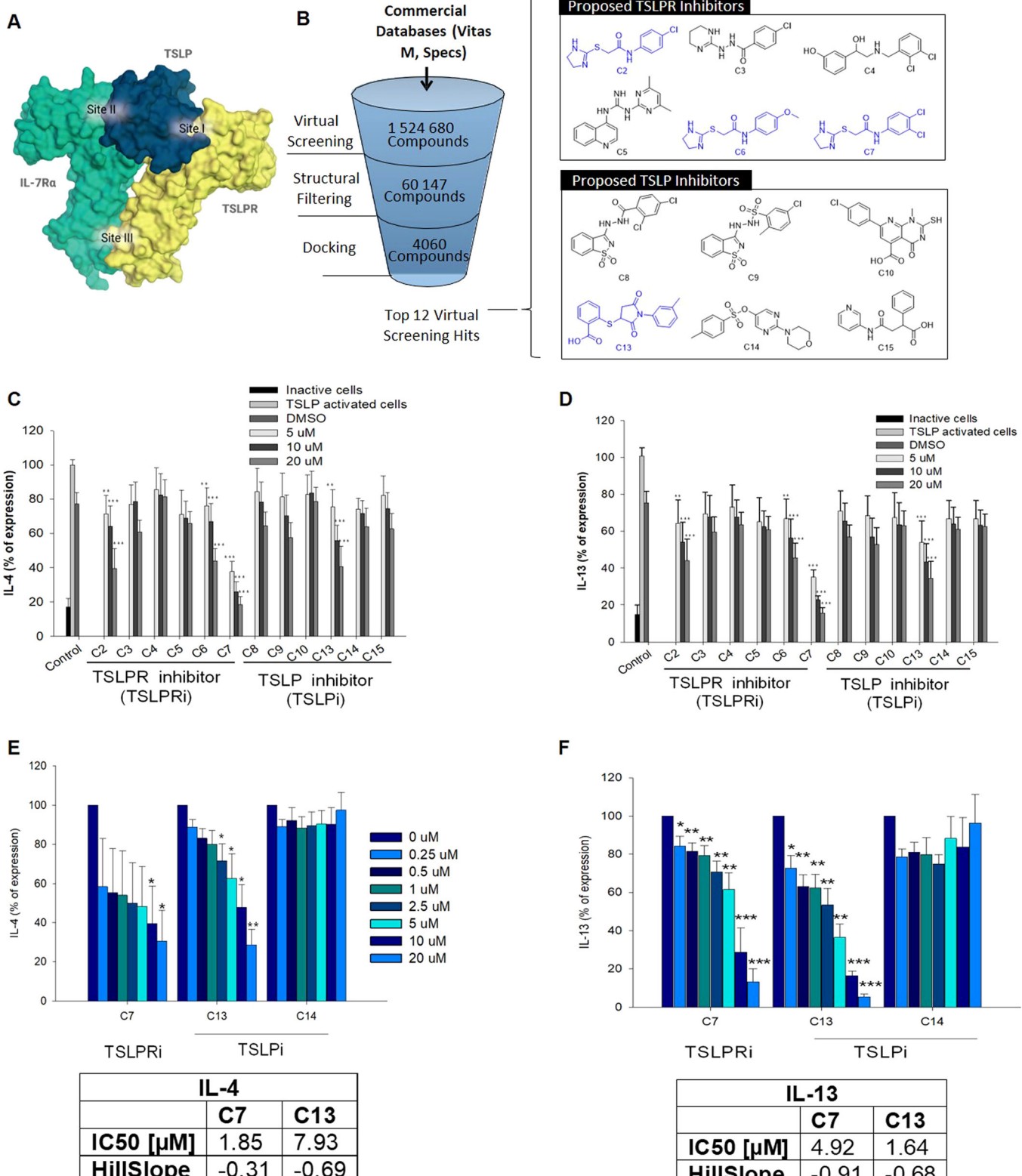

**Figure 1. In silico screening approach.**

(A) Schematic representation of TSLP, TSLPR and IL-7Rα forming a ternary complex; binding interfaces are indicated as site I, II, and III. (B) Overview of the virtual screening of potential small-molecule TSLP inhibitors and compound selection. A library of compounds from Specs and VitasM was screened using 3D pharmacophore models. (C, D) In vitro screening of twelve potential hit compounds in TSLP-activated HuT78 cells. Cells were treated with different concentrations of putative TSLP/TSLP receptor inhibitors for 36 h. IL-4 and IL-13 expression were detected with ELISA ($n = 3$). (E, F) Inhibition of Th2 cytokine secretion from human primary CD4 + T cells. TSLP-activated human primary CD4 + T cells from three donors were treated with different concentrations of compound 7, 13, and 14 for 36 h followed by quantification of IL-13 and IL-4 secretion via ELISA ($n = 3$). $IC_{50}$ values [μM] and slope were determined using GraphPad Prism software. Data information: All data are presented as mean ± SEM. $n$ represents the number of biological replicates. Statistical analysis was performed using Student's $t$ test, *$P \le 0.05$, **$P \le 0.01$, ***$P \le 0.001$. Source data are available online for this figure.

molecules were screened, focusing on the pre-defined key interactions and their fit in the cavity.

Eventually, twelve compounds of diverse chemical structures were selected for initial biological testing in the human TSLPR-expressing T-cell line HuT78 (Takahashi et al, 2016). First, the optimal experimental setup was determined showing 36 h as the ideal time for TSLP-mediated activation (Appendix Fig. S1). We further investigated the suitability of two activation approaches, PMA-ionomycin and CD3/CD28, the latter mimicking dendritic cell-mediated activation. Although both activation methods resulted in similar cytokine release patterns, PMA-ionomycin was selected for screening as it triggered higher cytokine levels overall (Appendix Fig. S2A).

Of the 12 tested compounds, C2, C6, C7, and C13 (Fig. 1C,D) significantly reduced TSLP-triggered IL-4 and IL-13 release in a concentration-dependent manner. C2, C6 and C7 have highly similar chemical structures, with subtle differences in their substitution patterns around their benzene ring, with the 3,4-dichlorobenzene functionality of C7 demonstrating the highest activity. Thus, C7 and C13 were further evaluated in primary human CD4 + T cells yielding equally strong inhibitory effects at 20 μM concentrations (Fig. 1E,F; Appendix Fig. S2B) while inducing no major cytotoxicity in primary human keratinocytes and fibroblasts (Fig. 2A–C).

The proposed binding mode of C7, designed to bind TSLPR, resembles the part of TSLP in *site I*. The nitrogen atoms present in the imidazole ring mimic the interaction formed by Arg153 of TSLP with Val193 of TSLPR thus anchoring the molecule in the interdomain elbow region of TSLPR. In addition, the dichlorophenyl moiety forms hydrophobic interactions with Leu39, Tyr143, Tyr194, and Val114 in our docking experiments thus suggesting further stabilization of the surmised binding conformation (Fig. 2D). The proposed binding mode of C13, designed to bind TSLP, is characterized by charge interactions and a hydrogen bonding network of the carboxylate moiety with Arg150 and Arg153 of TSLP, while the methylphenyl ring is interacting with Val67 and Leu147. In addition, there is a pi-cation interaction with Arg153 (Fig. 2D).

To initiate structure-activity relationship (SAR) studies with these top inhibitor scaffolds, we utilized the National Cancer Institute (USA) – Discovery Therapeutics Program (NCI-DTP) library to search for chemical analogs of C7 and C13. Substructure and similarity searches were performed using InstantJChem (ChemAxon), identifying 811 chemical analogs. These compounds were prepared for docking into the TSLP–TSLPR interaction interface using GLIDE (Schrödinger). Eighty-four top-scoring structural analogs of C7 and C13 were selected and tested in vitro using HuT78 cell line (Appendix Table S1). Some of these

compounds incorporated features of both C7 and C13 and retained some of their IL-4 and IL-13 inhibitory activity. None of the compounds from the NCI-DTP library outperformed C7 and C13, however, these data were useful in guiding design efforts focused on improving the activity of these validated hit compounds.

## Hit-to-lead development of novel TSLP inhibitors

Using inspiration from C7, C13, and the NCI-DTP compounds, a series of analogs were produced incorporating a diverse range of chemical functionalities. Specific efforts were made to explore the features outlined in Fig. 3A,B, modifying substituents on the aniline, rigidifying the core, exploring the impact of heterocycle substitutions, and incorporating ring systems with hydrogen bonding capacity. Notably, some of the most active NCI-DTP compounds did not possess a terminal aromatic ring system and instead possessed a carbamoylsulfanyl moiety (i.e., NSC13363 and NSC13350). Sixteen diverse compounds were synthesized and again screened for their efficacy to block TSLP-induced IL-4 and IL-13 release in Hut78 cells. From these sixteen compounds, BP79 most effectively blocked IL-13 (80% inhibition at 20 μM) and IL-4 (60% inhibition at 20 μM) release in a concentration-dependent manner (Fig. 3C,D). Combining features of C7 and C13 (i.e., BP86) led to a decrease in potency compared to both C7 and C13, supporting that these compounds are not likely to act as binders of both TSLP and TSLPR. Replacement of the sulfur atom with other heteroatoms appears to be reasonably tolerated, yet the most profound discovery was that the terminal dihydroimidazole or aromatic moiety was not required for potent inhibitory activity, as observed with BP79 (Fig. 3G; Appendix Tables S2–S7). This series also contained BP84, which was previously reported as an inhibitor of the TSLP signaling pathway, however in our hands this compound did not affect TSLP-induced expression of IL-4 and IL-13 in a statistically significant fashion.

Together with moderately active (BP96) and non-active compounds (BP75, BP80), BP79 was assessed in human primary CD4 + T cells to investigate inhibition of TSLP-mediated IL-4 and IL-13 expression (Fig. 3E,F). Again, BP79 demonstrated potent inhibition of IL-4 and IL-13 expression, BP96 gave a moderate level of inhibition, and the inactive compounds showed no inhibition of cytokine expression. This was further supported in dose–response studies. Here, BP79 inhibited TSLP-induced IL-4/IL-13 expression in primary CD4 + T cells with even stronger inhibitory effects (≥80% inhibition) at 10 μM and 20 μM concentrations (Fig. 4A,B). No significant cytotoxicity was observed at 20 μM in primary human skin cells and T cells (viability ≥80%; Fig. 3C–E), indicating that BP79 is well tolerated by primary cells. As an additional positive effect, the TSLP-induced T-cell hyperproliferation was abolished following BP79 treatment (Fig. 4F).

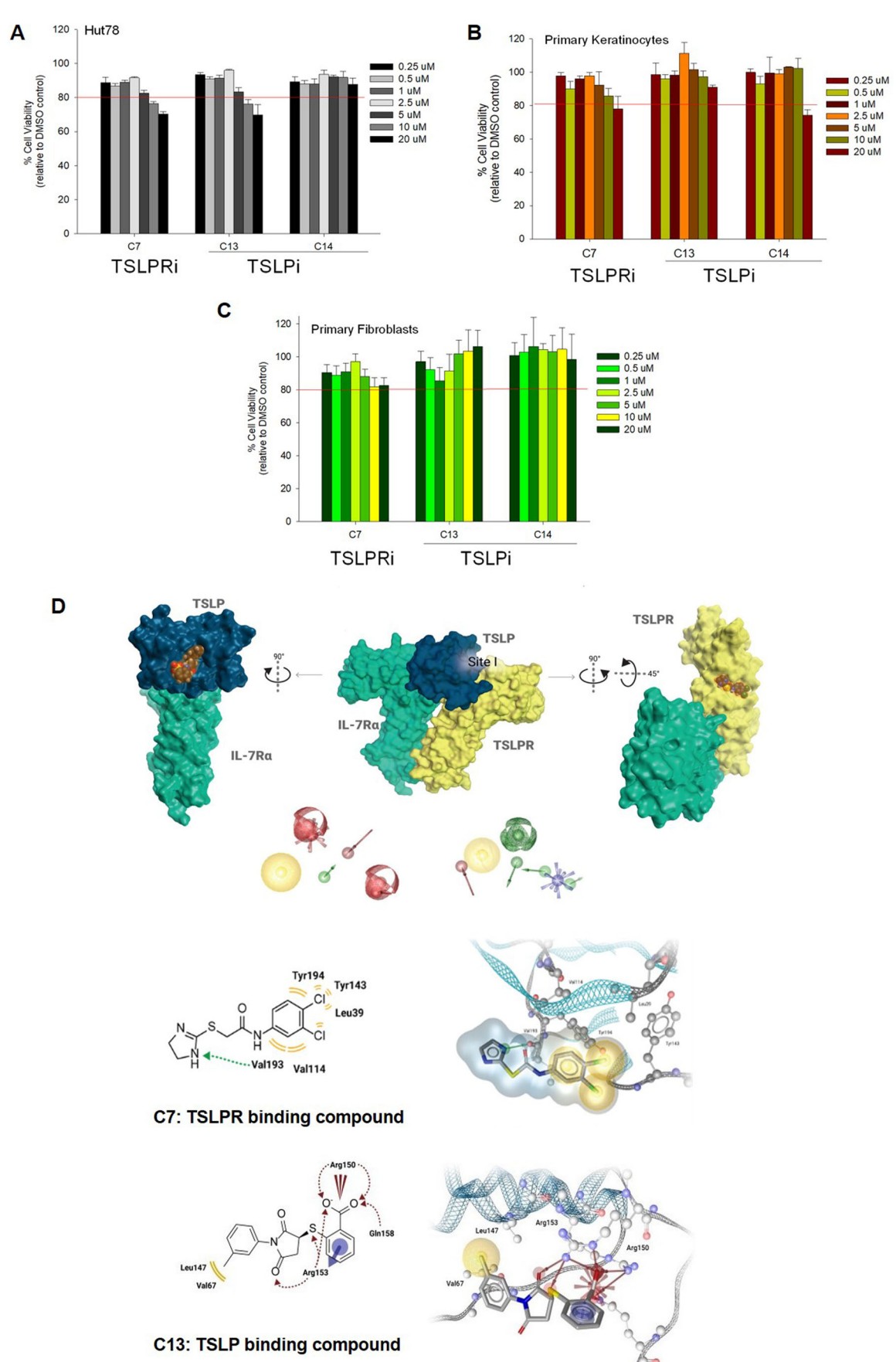

C7: TSLPR binding compound

C13: TSLP binding compound

◄  **Figure 2.  Biocompatibility and proposed binding modes.**

(A–C) Cell viability: Primary human keratinocytes, dermal fibroblasts, and HuT78 cells were treated with different concentrations of compound C7, C13, and C14. After 24 h, cell viability was determined by MTT assay ($n = 3$). (D) Proposed binding modes: The proposed binding mode of TSLPR-targeting C7 resembles the part of TSLP in site I. The nitrogen atoms present in in the imidazole ring mimic the interaction formed by Arg153 of TSLP with Val193 of TSLPR thus anchoring the molecule in the interdomain elbow region of TSLPR. Additionally, the dichlorophenyl moiety forms hydrophobic interactions with Leu39, Tyr143, Tyr194, and Val114 in our docking experiments, thus, suggesting further stabilization of the surmised binding conformation. The proposed binding mode of TSLP-targeting C13 is characterized by charge interactions and a hydrogen bonding network of the carboxylate moiety with Arg150 and Arg153 of TSLP, while the methylphenyl ring is interacting with Val67 and Leu147. In addition, there is a pi-cation interaction with Arg153. Data information: All data are presented as mean ± SEM. $n$ represents the number of biological replicates. Statistical analysis was performed using Student's $t$ test, $*P \leq 0.05$, $**P \leq 0.01$, $***P \leq 0.001$. Source data are available online for this figure.

BP79 contains a potentially reactive chemical moiety with an α,β-unsaturated carbonyl group. To investigate if this functionality may impart chemical reactivity/instability on this scaffold, we produced additional BP79 analogs that incorporated this same moiety. In total, 28 close structural analogs were produced and screened for their ability to suppress IL-4 and IL-13 expression in Hut78 cells at 1 µM concentration (Fig. EV1). Noting that BP243, 255, 262, 265, and 269 all had negligible inhibition of IL-13 expression, suggests that the presence of the α,β-unsaturated carbonyl is not the sole reason for the inhibitory activity of this series of compounds. Moreover, these screening results were supported by subsequent dose–response experiments using a diverse set of BP79 analogs in CD4 + T cells, where BP265 remained inactive, yet other compounds demonstrated dose-dependent inhibition of IL-4 and IL-13 expression. While several of these analogs were quite potent, none surpassed the potency of BP79 which inhibited both IL-4 and IL-13 expression to below 50% of controls upon 1 µM treatment (Fig. EV2). While it is possible that BP79 and other analogs may exert their inhibitory activity through a covalent inhibition mechanism, these experiments highlight that the activity is unlikely to be an artifact of chemical reactivity, and point to non-covalent interactions as the key role in the TSLPR inhibitory activity.

Next, we assessed the impact of BP79 on TSLP signaling. TSLP activates JAK1 and JAK2 and subsequently phosphorylates down-stream signaling proteins such as STAT3, STAT5 and STAT6. To identify the secondary messengers that stimulate Th2 cytokine secretion in primary human CD4 + T cells, TSLP-activated T cells were treated with inhibitors of JAK1/2 (ruxolinitib (Sada et al, 2021) and AZD1870 (Hedvat et al, 2009)), STAT3/5 (SH-4-54) (Haftchenary et al, 2013), STAT5 (573108) (Müller et al, 2008), and STAT6 (AS1517499) (Chiba et al, 2009) inhibitors showing that targeting the JAK-STAT pathway impairs IL-4 and IL-13 secretion (Appendix Fig. S3). In particular, the inhibition of JAK1/2, and STAT6 effectively downregulated Th2 cytokine secretion. In addition, the STAT3/5 dual inhibitor suppressed IL-4 and IL-13 secretion. However, STAT5 inhibition only modestly decreased Th2 cytokine secretion and primarily at rather high concentrations (100 µM). In keratinocytes, TSLP activates STAT3, thus, down-regulating the skin barrier protein filaggrin while upregulating TSLP expression (Kim et al, 2013; Dai et al, 2022). Interestingly, BP79 treatment blocked STAT3 and STAT6 phosphorylation in keratinocytes and CD4 + T cells (Fig. 4G,H), respectively, suggesting that its inhibitory effect on IL-4 and IL-13 release indeed results from an inhibition of TSLP signaling. Similar immune-dampening effects were observed in primary, human dendritic cells in which BP79 treatment abolished CCL17 expression and OX-40L; the latter being a critical co-stimulator for dendritic cell: T-cell interations

(Chen et al, 1999). Consequentially, IL-13 release was significantly reduced from T cells that were co-cultivated with BP79-treated dendritic cells (Fig. 4I–K).

## BP79 abolishes TSLPR complex formation without broad-spectrum kinase inhibitory effects

To verify the interaction of BP79 with TSLPR, we next investigated if BP79 blocks TSLP-mediated ternary complex formation using a proximity ligation assay (PLA). To visualize TSLPR/ IL-7Rα co-localization in primary human keratinocytes, TSLPR and IL-7Rα-specific antibodies were used to visualize the receptor complex formation following TSLP stimulation. Strikingly, BP79 treatment strongly inhibited TSLP-mediated receptor complex formation showing ≥ ten-fold inhibition compared to the untreated control (Fig. 5A). The non-active compound BP75 did not prevent the complex formation (Fig. EV3).

To further confirm the binding between BP79 and TSLPR, a western blot-based in vitro thermal shift assay on recombinant TSLPR was performed in the presence of BP79. While the TSLPR band was hardly detectable at 37 °C, pre-treatment with 20 µM BP79 stabilized the receptor up to 45 °C (Fig. 5B), indicative of a direct interaction between BP79 and TSLPR.

Finally, to evaluate if BP79 decreases the cytokine expression through direct binding of upstream kinases such as JAK1/2 or other kinase targets, BP79 was screened against 97 kinase targets using the KINOMEscan™ platform (radioligand displacement assay performed by Eurofins Discovery Services). BP79 demonstrated moderate binding of some kinases at 10 µM. Notably, MAPKAPK2 and MARK3 showed ≥50% indicative of additional kinase targets. There were limited inhibitory effects against JAK2, JAK3 and TYK2, which are part of the canonical JAK-STAT pathway that regulates STAT activation and inflammatory cytokine expression. While BP79 does not appear to be a broad-spectrum kinase inhibitor, these data cannot entirely rule out possible polypharmacologic effects associated with this compound (Fig. 5C). However, if BP79's activity was solely due to non-specific reactivity with multiple protein targets, a much more promiscuous kinase profile would be expected, supporting that its inhibitory effects are indeed due to selective inhibition of TSLP–TSLPR interactions.

## BP79 efficiently penetrates human skin and inhibits TSLP-mediated inflammation in an atopic-like skin disease model

Efficient skin absorption is a prerequisite for topical application and is governed by the physicochemical properties of the compounds. As such, we have defined key criteria for the initial

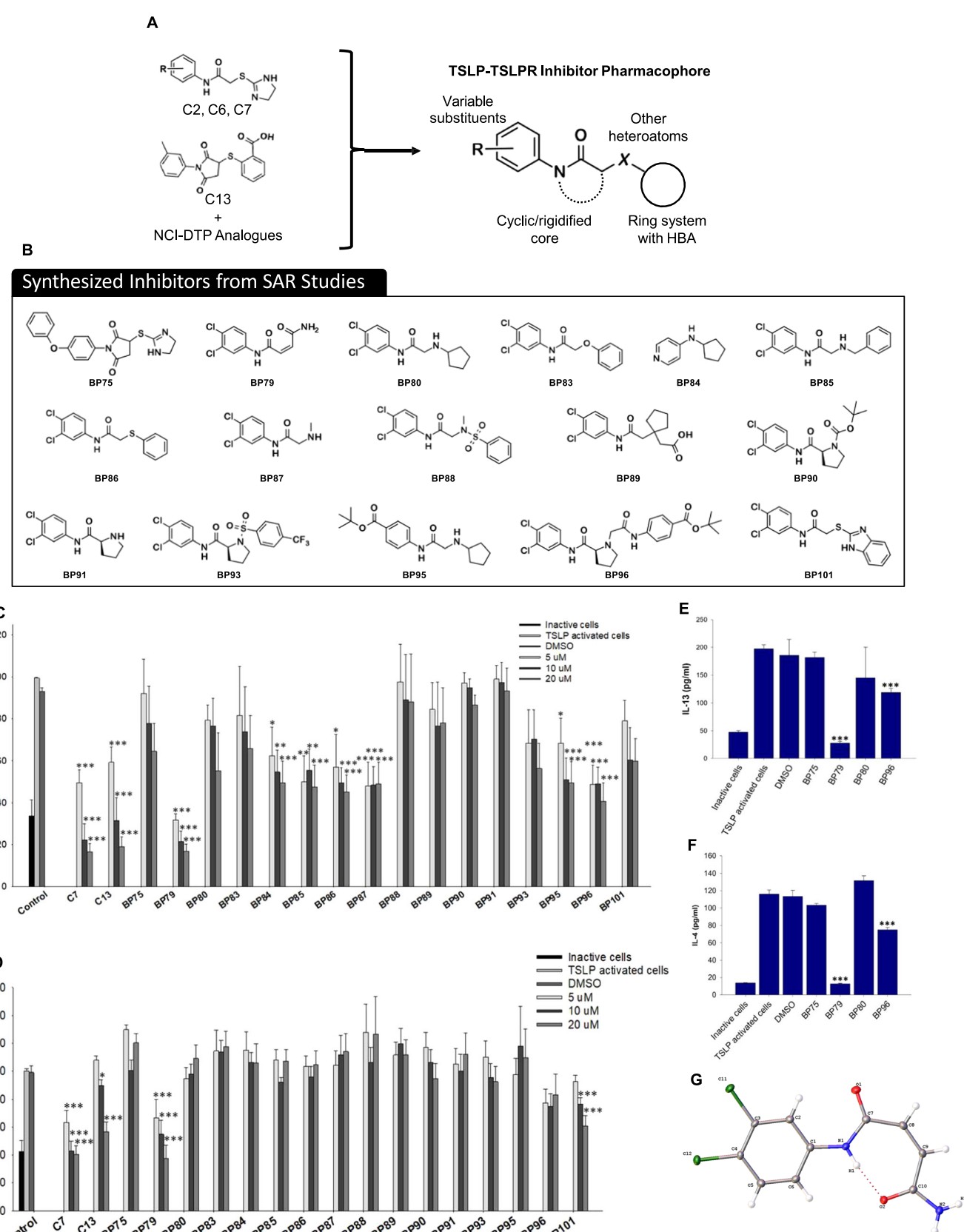

◀ **Figure 3.  Design of new TSLP inhibitors.**

(A) incorporating features of potent compounds from the initial screening campaign and hits from the NCI-DTP library. (B) Compounds were synthesized to optimize TSLPR inhibitory activity. (C, D) Inhibition of IL-13 and IL-4 secretion from Hut78 cells determined by ELISA. TSLP-activated HuT78 cells were treated with different concentrations of 16 newly synthesized TSLP inhibitors for 36 h ($n = 4$). (E, F) TSLP-activated human primary CD4 + T cells were treated with 20 μM of four promising compounds for 36 h. Secretion of IL-4 and IL-13 were determined using ELISA ($n = 3$). (G) Crystal structure of lead compound BP79, confirming the alkene substituents are in the cis-orientation and an intramolecular hydrogen bond between the aniline nitrogen atom and distal carbonyl oxygen confers rigidity to the overall structure. Data information: All data are presented as mean ± SEM. $n$ represents the number of biological replicates. Statistical analysis was performed using Student's $t$ test, *$P \leq 0.05$, **$P \leq 0.01$, ***$P \leq 0.001$. Source data are available online for this figure.

in silico work such as a molecular weight ≤800 Da and moderate lipophilicity (logP 1–3). As BP79 fulfills these criteria (Mw = 259.01 Da and calculated logP = 1.79 (MarvinSketch 23.1, ChemAxon)), it was not surprising that we indeed observed efficient skin absorption indicated by BP79 skin permeation after 8 h and even more pronounced 24 h (Fig. 6A).

We next utilized a previously established and extensively characterized atopic dermatitis-like 3D skin disease model to test BP79 effects in tissue models of high clinical biomimicry. This skin disease model is based on filaggrin gene knockdown (Fig. 6B), while the inflammatory phenotype is induced by adding IL-4 and IL-13 (Hönzke et al, 2016). Previous studies have verified its atopic-like characteristics including impaired skin barrier function, high TSLP expression, facilitated T-cell migration, and impaired skin surface pH regulation (Vávrová et al, 2014; Hönzke et al, 2016; Wallmeyer et al, 2017). In line with previous studies, TSLP expression significantly increased in the disease models. The addition of activated CD4 + T cells further enhanced TSLP secretion and an array of other, atopy-relevant markers such as IL-5, IL-2, IL-4, IL-9, IL-13, and IFNγ (Fig. 6B,C) compared to skin-healthy samples. Importantly, the topical application of BP79 highly significantly reduced the expression of these markers, IL-22, and TNFα, indicative of strong anti-inflammatory effects.

## BP79 exerts potent anti-inflammatory effects in a complex, human-based atopic diseases-on-a-chip model: a novel drug discovery platform

Due to limited functional cross-reactivity between human and murine TSLP, and distinct inter-species-related differences, further preclinical testing of BP79 in an in vivo model proved extremely difficult.

To overcome this limitation, we developed a microfluidic two-organ chip model to study the efficacy and safety of BP79 in a complex, human-based setup. This platform contained an atopic-like skin disease model that was co-cultivated with a 3D bronchial epithelial tissue model on a dynamic organ chip platform that ensured media circulation and flow from the skin to the healthy lung tissue (Fig. 7A,B). Activated CD4 + T cells were added to the circuit due to their role as direct TSLP effector cells.

Initially, we focused on model establishment and validation. Importantly, no histological changes were observed in the bronchial epithelial models after co-cultivation with normal skin models. However, the atopic-like skin model showed a significantly thickened epidermal layer and strong parakeratosis and exerted detrimental effects on the bronchial epithelial models that was characterized by impaired tissue cohesiveness and pronounced cell shedding (Fig. 7C). The culture media of the atopic diseases chip contained significantly higher levels of classic Th2-derived

cytokines like IL-4, IL-13, as well as TSLP and periostin (Fig. 7D). Except for fluctuations due to media changes, LDH and glucose levels remained constant indicative of a homeostatic setup (Appendix Fig. S4).

Next, BP79 was topically applied onto the atopic-like skin disease models once every 24 h over 4 days. As a consequence, a distinct downregulation of key inflammation markers such as secreted TSLP, periostin, and IL-13 to levels observed in healthy chip setups were observed (Fig. 7D). BP79 treatment also exerted beneficial effects on the epithelial barriers, as exemplified by increased filaggrin and decreased TSLP mRNA and protein expression in the skin and lungs (Figs. 7E & 8A,B; Appendix Fig. S5). On mRNA level, a significant decrease of KRT5 and KRT14, markers for non-differentiated, proliferation-competent cells, was also noted following BP79 and tacrolimus treatment, whereas for BP79 also a trend toward increased KRT10 expression was observed, indicative for improved differentiation. At the same time, T-cell infiltration into the skin models was efficiently blocked which occurred at high levels in the untreated atopic conditions (Fig. 8B,C).

To determine the ability of our atopic diseases chip to reflect drug-specific effects, we included the clinically approved calcineurin inhibitor tacrolimus as a reference. While tacrolimus also significantly suppressed IL-4 and IL-13 secretion, it did not modulate the expression or secretion of periostin and TSLP. Similarly, it did neither prevent T-cell migration into the disease model nor increased filaggrin expression (Fig. 8B,C). For the bronchial epithelial models, increased TSLP expression was noted after co-cultivation with untreated skin disease models which was less pronounced after BP79 treatment. Similarly, a more pronounced CD4$^+$ T-cell infiltration was observed in bronchial models co-cultivated with the untreated disease models which was reduced after BP79 treatment although less pronounced compared to skin (Appendix Fig. S6).

To gain additional insights on the effect of BP79 treatment on both skin and lung tissue, both tissues (treated and untreated) were subjected to transcriptomic analysis with RNA-Seq. Subsequently, we compared treated *versus* untreated skin and lung samples. For atopic-like and normal skin models, we observed a large number of differentially expressed genes, indicating a strong treatment effect. At FDR < 0.01, there were 1080 genes with at least twofold difference in treated *versus* untreated diseased skin models (Appendix Fig. S7) substantiating the presence of the disease phenotype, and the anti-inflammatory effect of BP79. On gene level, the topical application of BP79 onto diseased skin significantly changed protein synthesis and cell metabolism (Appendix Table S8; Fig. 8D). Further, we found that the transcription profiles of treated diseased skin were more similar to healthy skin than untreated diseased skin. For example, at FDR < 0.01 and absolute log2 FC > 2, there were 1080 DEGs between treated and

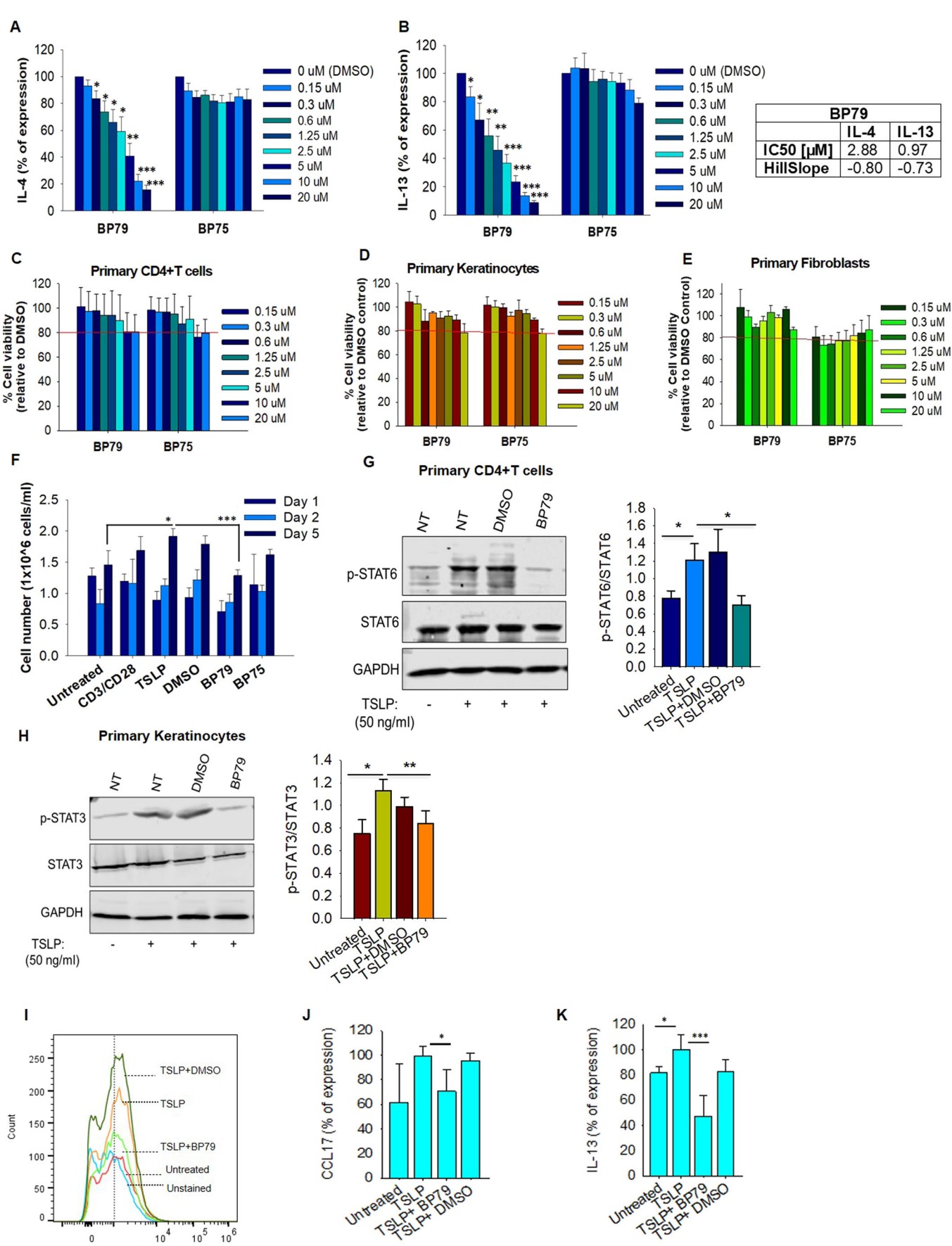

**Figure 4. Functional validation of BP79.**

(A, B) TSLP-activated primary CD4 + T cells were treated with different concentrations of the TSLP inhibitors BP75 and BP79 for 36 h, respectively. IL-13 and IL-4 secretion was quantified by ELISA ($n = 6$), mean ± SEM. IC$_{50}$ values [μM] and slope were determined using GraphPad Prism software. (C–E) Cytotoxicity of BP79 in primary human keratinocytes ($n = 4$), fibroblasts ($n = 5$), and CD4 + T cells ($n = 5$) was determined by MTT assay. (F) Inhibition of TSLP-mediated T-cell hyperproliferation. Primary CD4 + T cells were treated with TSLP inhibitors (or DMSO) and stimulated with TSLP for 1, 2 and 5 days. Cell proliferation was determined using a cell counter; $n = 3$. (G) Primary human CD4 + T cells and (H) keratinocytes were treated with BP79 and stimulated with TSLP. BP79 inhibited STAT6 and STAT3 activation in primary CD4 + T cells and primary keratinocytes, respectively ($n = 3$). The band intensities were quantified using ImageJ ($n = 3$). Phospho-Stat6 and Phospho-Stat3 expressions were normalized to Stat6 and Stat3 expressions, respectively. (I) Primary human myeloid dendritic cells were treated with BP79 and stimulated with TSLP for 24 h. OX-40L expression was detected using flow cytometry. (J) Primary human myeloid dendritic cells from three donors were treated with BP79 and stimulated with TSLP for 24 h. CCL17 expression was detected using ELISA ($n = 3$). (K) Inhibition of dendritic cell-mediated Th2 cell differentiation. IL-13 secretion was determined using ELISA ($n = 3$). Data information: All data are presented as mean ± SEM. $n$ represents the number of biological replicates. Statistical analysis was performed using Student's $t$ test, *$P \leq 0.05$, **$P \leq 0.01$, ***$P \leq 0.001$. Source data are available online for this figure.

untreated diseased skin, 809 DEGs between treated diseased skin and healthy skin, and as much as 3331 DEGs between untreated diseased skin and healthy skin. Also, the Spearman correlation coefficients of the average gene expression were highest when comparing healthy skin with treated diseased skin. A discordance/concordance plot and heat map of the top 25 genes comparing "healthy vs treated", "healthy vs untreated", and "treated vs untreated" further corroborates this finding (Appendix Figs. S8 and S9). Interestingly, beneficial effects on the lung tissue were noted following BP79 administration onto the skin model. Pathway enrichment analysis for treated vs. untreated lungs showed enrichment in the KEGG and REACTOME aminoacyl tRNA biosynthesis pathway. Another enriched REACTOME pathway was sodium proton exchangers ($P = 0.0048$, Fig. 8E, Appendix Table S9).

# Discussion

While the scientific community has made great strides in understanding the pathomechanism of atopic diseases, comparably little progress has been made in the context of treating and more so preventing atopic diseases.

Calcineurin inhibitors and glucocorticoids remain the first-line medications for AD, which, however, are accompanied by dose-limiting side effects including skin atrophy and the risk of malignancies (Niculet et al, 2020; Ume et al, 2020). In addition, the available treatment options often do not achieve sufficient disease control in many patients with high disease activity. For instance, every 5[th] child affected by AD typically try ≥ 15 different medications to sufficiently control the disease (ESC, 2017).

Recently, several new drugs such as JAK inhibitors have been approved which expand the treatment options. Similarly, biologics such as the anti-IL-4/IL-13 antibody dupilumab are now available for the treatment of moderate-to-severe AD. While clinical data show favorable effectiveness and a good safety profile, still a certain number of therapy-resistant patients remain. Also, biologics are generally associated with high therapy costs of ~$30,000 US per year (Kamata and Tada, 2023). This renders the development of safe and effective small-molecule drugs especially for moderate-to-severe atopic dermatitis urgent (ESC, 2017) which holds particularly true for children, which are most frequently affected.

TSLP has been recognized as a central driver of atopic diseases and, thus, has received increasing attention as a therapeutic target. Consequentially, anti-TSLP antibodies such as tezepelumab have been developed, acknowledging the key role of TSLP in the pathogenesis of atopic diseases. Interestingly, while being approved

for severe asthma, disappointing results were obtained for clinical studies in AD. Here, tezepelumab did not reach the targeted efficacy in dampening the inflammation and itch (Simpson et al, 2019). This was surprising given the central role of TSLP in AD pathogenesis and further studies are needed to better understand this outcome.

While biologics such as tezepelumab have very high target specificity, and, thus fewer side effects, they are accompanied by high costs and complex storage and shipping conditions. Further, topical drug administration offers distinct advantages over the required systemic administration of antibodies such as localized drug delivery and improved patient acceptance. Also, primary TSLP-mediated sensitization for allergic asthma happens in the skin in the context of AD (Han et al, 2012). Thus, a topical application of TSLP inhibitors on infant skin early on could potentially even avert the onset of disease progression and, thus, the atopic march.

Hence, building onto that and aiming to close current therapeutic gaps, we leveraged in silico approaches, rational drug design, and the application of state-of-the-art chemical biology tools to develop small-molecule TSLP inhibitors (Fig. 1).

Following several screening and optimization rounds, BP79 emerged as our lead candidate due to strong inhibitory effects in the micromolar range in relevant cell types while showing good cytocompatibility (Fig. 3). Its target engagement was confirmed by a range of methods including proximity ligation (PLA) and thermal shift assays (Fig. 4). Other classic tools to verify target engagement such as surface plasmon resonance (SPR) measurements did not produce conclusive results. This may be due to the following reasons: (1) TSLPR and IL-7R are transmembrane receptors. While recombinant proteins are commercially available, their quality, purity, and functionality are highly variable. It is possible that they are not folded correctly, thus preventing to assess the binding of small molecules. Such recombinant proteins are typically stabilized by His- or Fc-tags, which may also impact the protein structure, thus not resembling the native state. Nevertheless, the presented combination of in vitro thermal shift assay, proximity ligation assay and thorough analysis of additional chemical inhibitors provide strong supporting evidence that BP79's activity is due to disrupting TSLP–TSLPR interactions and subsequently blocking downstream signaling events.

We further demonstrate that BP79 is not a broad-spectrum kinase inhibitor, and importantly lacked activity against the JAKs, which are canonical mediators of TSLP-induced STAT signaling and inflammatory cytokine expression (Fig. 5). A PLA assay confirmed that BP79 disrupts the ternary receptor complex formation between TSLP/TSLPR/IL-7Rα, thus inhibiting downstream signaling. Moreover, we found that BP79 treatment

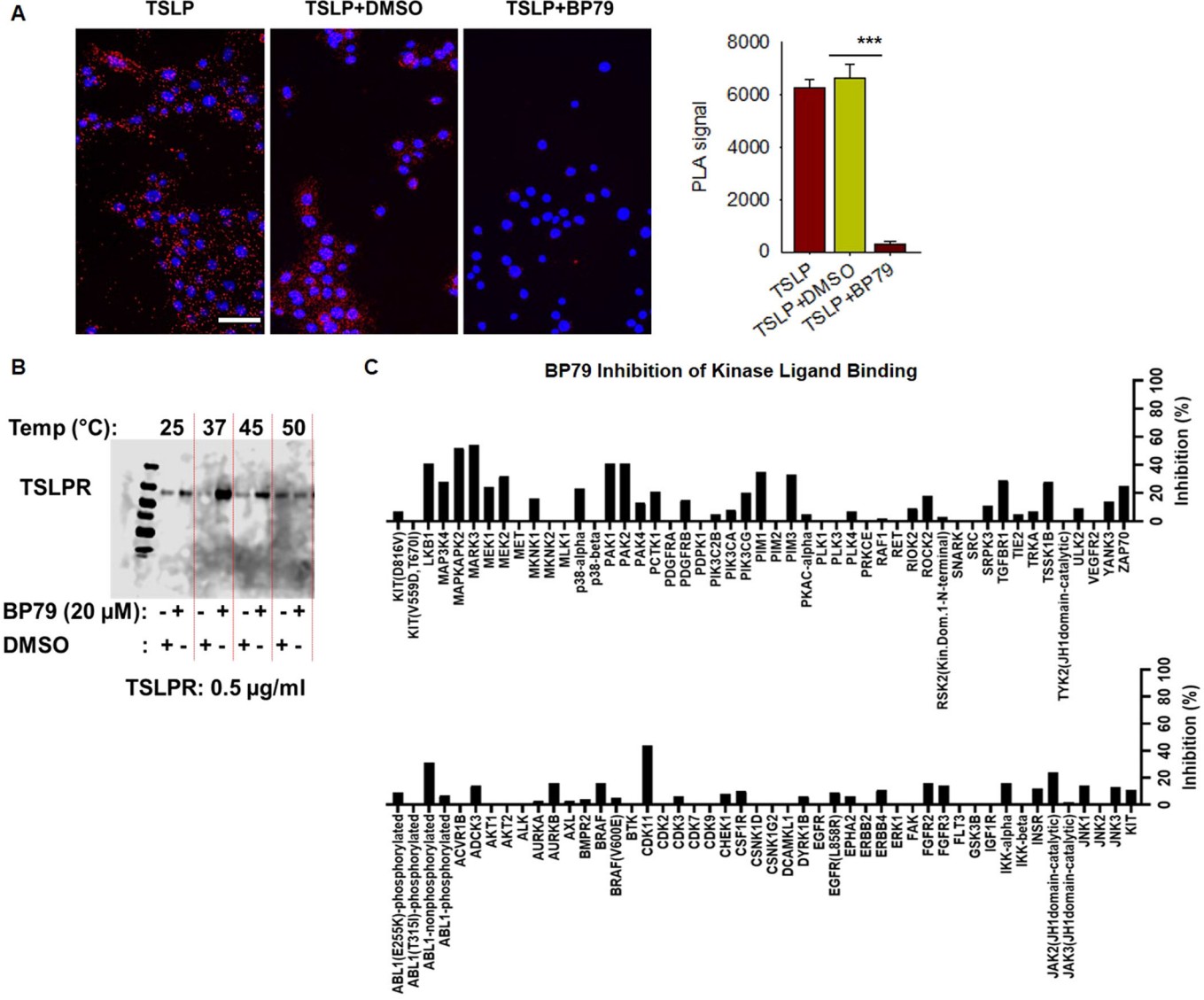

**Figure 5. Target engagement and specificity of BP79.**

(**A**) Proximity ligation assay (PLA): TSLP binds to TSLPR and forms a ternary complex with IL-7Rα at the cell surface initiating intracellular signaling. PLA was performed to verify the inhibition of TSLP-mediated ternary complex formation via BP79. Primary keratinocyte cells were treated with the TSLP inhibitors and stimulated with TSLP. Cells were fixed and the interaction between TSLPR and IL-7Rα was observed with PLA assay. BP79 inhibited the TSLP:TSLPR:IL-7Rα ternary complex formation. Images were counted using ImageJ ($n = 20$). Scale bar = 50 μm. (**B**) In vitro thermal shift assay. BP79 thermally stabilized TSLPR up to 45 °C. (**C**) To determine if BP79 decreases the cytokine expression through direct binding of upstream kinases such as JAK1/2 or other kinase targets, BP79 was screened against 97 kinase targets using the KINOMEscan™ platform (radioligand displacement assay performed by Eurofins Discovery Services). Data information: All data are presented as mean ± SEM. $n$ represents the number of biological replicates. Statistical analysis was performed using Student's $t$ test, *$P ≤ 0.05$, **$P ≤ 0.01$, ***$P ≤ 0.001$. Source data are available online for this figure.

abrogated TSLP-mediated proliferation of CD4 + T cells and effectively dampened dendritic cell-mediated immune reactions (Figs. 4 and 5). Previous work from our lab also demonstrated that TSLP stimulates T-cell migration even in the absence of dendritic cells (Wallmeyer et al, 2017), which consequentially is reduced when TSLP-mediated effects are abrogated.

Aiming to study the pharmacological efficacy and safety in a complex disease model, we opted to develop a human-based atopic drug discovery platform leveraging an organ-on-chip technology (Fig. 7). This decision was driven by the fact that conducting these

studies in animal models proved difficult. While murine and human TSLP operate in a similar manner (Verstraete et al, 2014; Verstraete et al, 2017), they only share 43% and 35% homology for TSLPR, respectively, with no cross-reactivity between the species (Savino and Izraeli, 2016). Therefore, to the best of our knowledge, there is no animal model available to accurately mimic human TSLP-mediated Th2 inflammation in atopic diseases. Our TSLP inhibitors have been specifically designed to block human TSLPR, rendering currently available mouse models unsuitable. In addition, atopic diseases do not spontaneously develop in most rodents and,

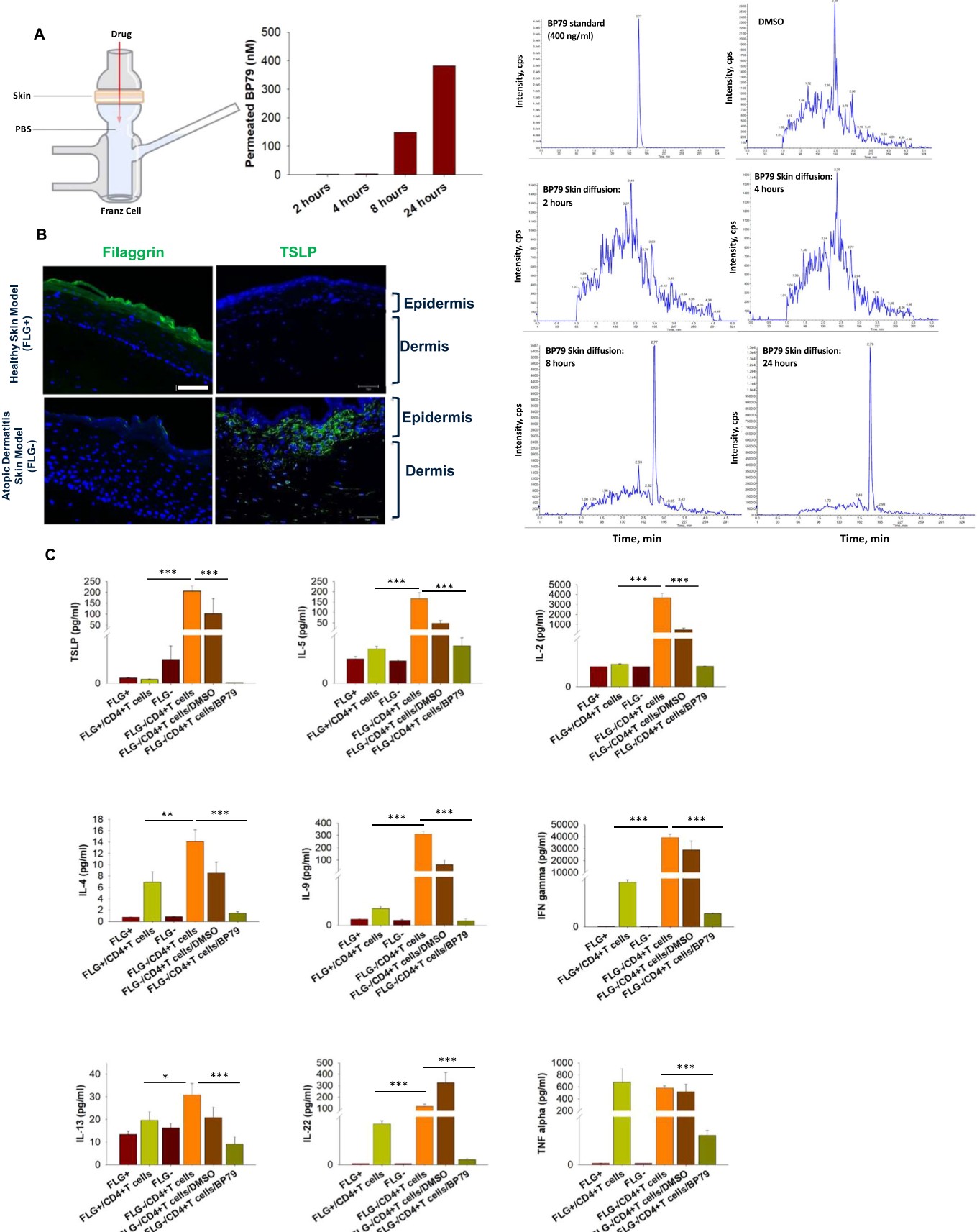

**Figure 6.  Efficacy of BP79 in human skin and 3D tissue model.**

(A) Skin permeation of BP79 was determined using a Franz cell setup followed by MS analysis showing BP79 permeation after 8 and 24 h, respectively. (B) Expression of filaggrin and TSLP in normal (FLG + ) and atopic-like (FLG-) skin models as determined by immunofluorescence staining. Scale bar = 25 μm. (C) Efficacy testing of BP79 in normal (FLG + ) and atopic-like (FLG−) skin models in the presence or absence of activated CD4+ cells emulating atopic-like disease conditions in a human-based setup. BP79 or the DMSO vehicle control were topically applied over three consecutive days. Cytokine expression was then quantified using MSD U-PLEX Cytokine Array Assay kit (n = 3). Data information: All data are presented as mean ± SEM. n represents the number of biological replicates. Statistical analysis was performed using Student's t test, *P ≤ 0.05, **P ≤ 0.01, ***P ≤ 0.001. Source data are available online for this figure.

thus, are usually highly artificially induced, with uncertain fidelity to the human situation (Löwa et al, 2018).

3D skin models and excised human skin provide a more realistic representation of the skin barrier and enables the evaluation of permeability and efficacy of BP79 in a more clinically relevant context. Further, complex human-based organ-on-a-chip setups have emerged as suitable alternatives to address this translational gap (Loewa et al, 2023). Since TSLP is involved in the progression of atopic diseases and also plays a critical role in the manifestation of allergic asthma, we developed an immunocompetent two-organ-on-chip setup as a drug discovery platform that has broad application potential for preclinical testing of anti-inflammatory drugs beyond small-molecule TSLP inhibitors.

The atopic diseases-on-chip platform reliably demonstrated the anti-inflammatory effects of BP79 and differentiated its mode of action from tacrolimus, a calcineurin inhibitor. The latter predominantly dampens inflammation through inhibitory effects on T-cell proliferation by blocking IL-2 production (Kraaijeveld et al, 2019) (Fig. 7).

Interestingly, pathway enrichment analysis following RNA-Seq of treated vs. untreated lung models showed enrichment in the KEGG and REACTOME aminoacyl tRNA biosynthesis pathway, which plays a critical role in immune regulations, chemoattraction for leukocytes, and contribute to the maturation, activation, and recruitment of immune cells (Fig. 8) (Nie et al, 2019). Other enriched REACTOME pathways included sodium proton exchangers (P = 0.0048), which regulate the water content of lung fluids. Some studies suggest that dysregulated airway hydration may promote airway inflammation (Wang and Ji, 2015). For instance, sodium proton exchanger regulatory factor 1, another enriched pathway, was shown to facilitate allergic airway inflammation in mice accompanied by increased immune cell infiltration, goblet cell hyperproliferation, increased Th2 cytokine expression (Kammala et al, 2021).

Taken together, our study reports the first putative small-molecule TSLP inhibitor that exerts potent anti-inflammatory effects in the low micromolar range and has a great potential to broaden the spectrum of topically applicable drugs for the treatment of atopic dermatitis and, maybe even the prevention of the progression of atopic diseases to other epithelial. Further, we present a complex, human-based drug discovery platform for atopic diseases which may close the current gap of human-based, translational preclinical models in the area of atopic diseases.

# Methods

## Cell culture

HuT78 cells were freshly sourced from the American Type Culture Collection (ATCC; male donor) and checked for mycoplasma contamination as appropriate. Primary human peripheral blood

mononuclear cells (PBMCs) and CD4 + T cells were isolated from healthy blood donor buffy coats (CREB approval 2019.023). Primary human dermal fibroblasts (FBs) and keratinocytes (KCs) were isolated from juvenile foreskin according to standard procedures (CREB approval #H19-03096). Written informed consent was obtained from all subjects in line with the Declaration of Helsinki and the Department of Health and Human Services Belmont report. Normal human lung fibroblasts were purchased from CELL applications (Cat# 506-05a). Normal human bronchial epithelial cells (NHBEs) were purchased from EPITHELIX (Cat# EP51AB).

PBMCs were isolated from buffy coat using Lymphoprep density gradient centrifugation (STEMCELL, Catalog # 07851) and CD4 T cells were separated from PBMCs using EasySep human CD4+ T-cell isolation kit (STEMCELL, Cat# 17912) following the manufacturer's instructions. HuT78, PBMCs and CD4+ T cells were cultured in RPMI-1640 Medium (Sigma, R8758) supplemented with 10% (v/v) of FBS (Sigma, F1051) and 1% (v/v) of penicillin–streptomycin (Sigma, P4333); media were changed every three days. Myeloid dendritic cells (mDC) were isolated from PBMCs using EasySep™ human myeloid DC enrichment kit (STEMCELL, Cat# 19061). Dendritic cells were cultured in human dendritic cell medium (Cell Applications, Cat#623-50). Dermal FBs and Normal human lung FBs were cultured in DMEM – high glucose (Sigma, D6429) supplemented with 10% (v/v) of fetal bovine serum (FBS) (Sigma, F1051) and 1% (v/v) of penicillin–streptomycin (Sigma, P4333), medium change was performed every two days. KCs were cultured with EpiLife medium (Gibco, MEPI500CA) supplemented with human keratinocyte growth supplement (Gibco, s-001-5). NHBEs were cultured with PhneumaCult-Ex medium (STEMCELL, Cat# 05008). Media were changed every other day, unless otherwise stated. All cells were maintained at 37 °C in a humidified 5% $CO_2$ incubator.

## Reagents and antibodies

Potential TSLP small-molecule inhibitors were obtained from VitasM, Specs and NCI-DTP libraries. Uncoated Human IL-13, IL-4, and TSLP ELISA Kits were obtained from Invitrogen (Cat# 88-7439-88, # 88-7046-88, # 88-7497-88), Human CCL17/TARC DuoSet ELISA (Catalog #: DY364) and Periostin ELISA kits (Cat# DY3548B) were purchased from R&D Systems and the U-PLEX human cytokine array assay kit from Meso Scale Discovery, Inc (Cat# K15067L-1). BCA protein assay kit was obtained from G Bioscience (Cat# 786-570). JAK1/2 inhibitor-Ruxolitinib (INCB018424) was purchased from Selleckchem. JAK2 inhibitor (AZD1480), Stat3/Stat5 inhibitor (SH-4-54) and Stat6 inhibitor (AS1517499) were purchased from Cayman Chemicals. Stat5 inhibitor (573108), human recombinant TSLP (Cat# SRP4896), phorbol-12-myristate-13-acetate (PMA, Cat# P1585), ionomycin (Cat# I0634), human anti-CD4 antibody (Cat# MABF419),

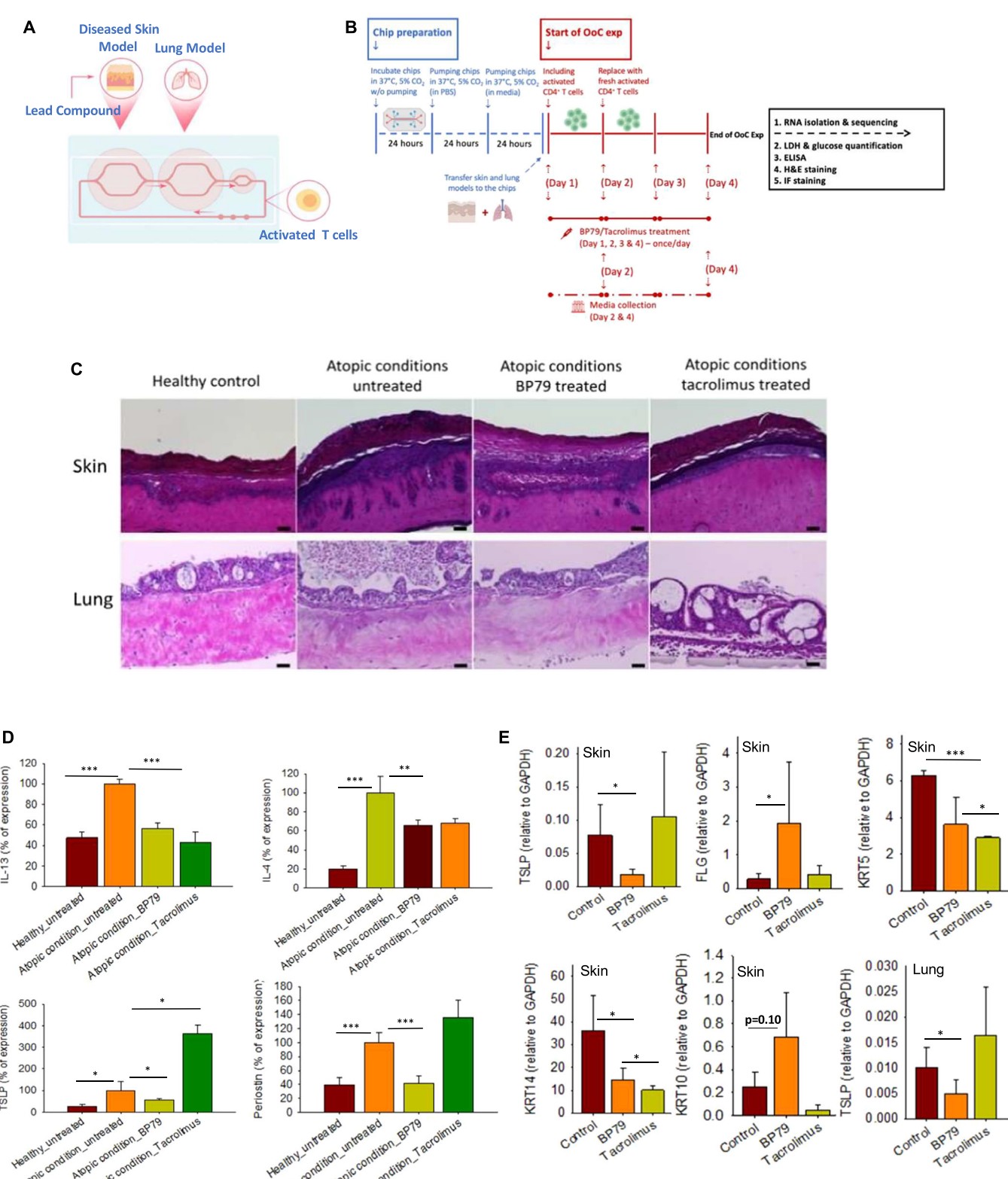

Thiazolyl Blue Tetrazolium Bromide (MTT, Cat# M2128), Proximity Ligation Assay Kit (Cat# DUO92103-1KT) were purchased from Sigma Aldrich. GAPDH (Cat# 2118S), p-STAT5 (Y694) (Cat# 9359S), STAT5 (Cat# 25656S), p-STAT3 (Y705) (Cat# 9131S), STAT3 (Cat# 4904), p-STAT6 (Y641) (Cat# 56554S), STAT6 (Cat# 5397S), TSLP (Cat# 97630S) were obtained from Cell Signaling Technology. TSLPR (Cat# PA547979), IL-7Rα (Cat#14-1278-82) and secondary antibodies (Cat# A21206, Cat# A28174) were obtained from Invitrogen. Filaggrin

**Figure 7. Development & utilization of an atopic diseases-organ chip for preclinical BP79 testing.**

(A, B) Experimental setup of the co-cultivation of normal or atopic-like skin models with 3D normal bronchial epithelial models in a microfluidic two-organ chip. BP79 and tacrolimus were topically applied onto the skin (disease) model. (C) H&E staining of co-cultivated skin (normal and diseased) and lung tissue models w/ or w/o topical application of BP79 or tacrolimus. (D) Topical application of BP79 significantly reduced IL-13, IL-4, TSLP and periostin expression as detected via ELISA. Scale bar = 50 μm. (E) Relative mRNA expression of TSLP, filaggrin, keratin 5, keratin 14, and keratin 10 in atopic-like skin and bronchial lung tissue models following topical treatment with BP79 and tacrolimus determined via quantitative RT-PCR (*n* = 3). GAPDH served as a housekeeping gene.

(Cat# ab814688), CK14 (Cat# ab7800), IVL (Cat# ab68) antibodies were obtained from Abcam. PE anti-human CD252 (OX-40L) antibody was purchased from Biolegend (Cat#326308). Lymphoprep (Catalog # 07851) and ImmunoCult Human CD3/CD28 T Cell Activator (Cat # 10991) were purchased from STEMCELL. PureLink™ RNA Mini Kit was obtained from Invitrogen (Cat# 12183018A). iScript cDNA Synthesis Kit and SsoAdvanced Universal SYBR Green Supermix were purchased from Bio-Rad (Cat# 1708891 and # 1725270).

## In silico screening of library compounds

### Protein preparation

All modeling studies were performed using the crystal structure of human TSLP in complex with TSLPR and IL-7Rα (PDB (Berman et al, 2000) code: 5j11 (Van Rompaey et al, 2017)). Since we were interested in designing ligands to disrupt the TSLP:TLSPR interface, we prepared chain A and B of this crystal structure by removing crystal waters and by assigning protonation states using the "Protonate3D" (Labute, 2009) application implemented in MOE v2019.0102 (2006).

Cavity detection and binding site analysis: After removing TSLP (PDB 5j11 (Van Rompaey et al, 2017), chain A) and TSLPR (PDB 5j11 (Van Rompaey et al, 2017), chain B), respectively, we performed a binding site analysis using "Site finder", a cavity detection program based on (Del Carpio et al, 1993), implemented in MOE v2019.0102 (2006). The analysis was focused on site I (Verstraete et al, 2014) at the interface between TSLP and TSLPR. Since the interaction of Arg150, Arg153 of TSLP with Asp92 of TSLPR is known to be crucial (Verstraete et al, 2014; Verstraete et al, 2017), only solutions including this interaction were considered.

### 3D pharmacophore-based virtual screening and database preparation

3D pharmacophores were developed using LigandScout 4.4. As no crystal structure with co-crystallized small molecule nor any small-molecule TSLP inhibitors were available, key interactions were manually placed at the interface for the side of TSLP or TSLPR, respectively. Both 3D pharmacophore models were used to screen 1,524,680 commercially available compounds from Specs (Delft, Netherlands) and VitasM (Hongkong, China). Before screening, both databases were standardized, assigned correct charges at pH 7.4 and counter ions were removed using an in-house standardization workflow using KNIME (Berthold et al, 2009) and RDKit (Landrum, 2006). LigandScout's (Wolber et al, 2006; Wolber and Langer, 2005) idbgen program was used to pre-calculate conformational models for all molecules. Virtual hits were filtered limiting the number of rotatable bonds, molecular weight, and matching features to the developed pharmacophore. All compounds with less than seven rotational bonds and greater than 260 Daltons

molecular weight with at least five matching chemical features were selected for docking studies.

### Molecular docking

To check whether virtual hits from 3D pharmacophore screening are actually able to fulfill the desired interaction pattern, a molecular docking step was included in the virtual screening workflow. Virtual hits were docked using GOLD v5.2 (Jones et al, 1997) with standard parameters and GoldScore (Verdonk et al, 2003) as a scoring function. Binding site residues were defined as all protein residues in 10 Å distance from the key residues Asp92 or Arg153 for TSLPR and TSLP, respectively. Resulting docking poses were minimized in their respective binding pockets using LigandScout's (Wolber et al, 2006; Wolber and Langer, 2005) MMFF94 (Halgren, 1996a; Halgren, 1996b) implementation.

For NCI-DTP compounds, chemical structures were imported into Maestro (Schrödinger) and prepared for docking using the ligprep function. TSLPR from PDB 5J11 was prepared for docking using standard workflows, which included preprocessing, removing water molecules, and energy minimization. A docking grid was set as a cube with 10 Å sides centered around TSLPR Asp92. Compounds were docked with extra precision and docking scores were ranked. Top-scoring compounds were acquired from the NCI; however, some compounds were not available at the time of these experiments.

## General chemical methods

Detailed schemes, chemical procedures, and analytical data are included in the supplementary materials. Reagents were sourced from Millipore Sigma (Burlington, MA, USA), Combi-blocks (San Diego, CA, USA) or AK Scientific (Union City, CA, USA) and used without modification. Anhydrous solvents were sourced from Sigma Millipore and used without further modifications. Compounds were purified using automated flash chromatography (Biotage Isolera, Uppsala, Sweden) with ACS grade solvents (hexanes, ethyl acetate, methanol, and dichloromethane) from Fisher Scientific (Waltham, MA, USA), or preparative HPLC (Agilent, Infinity II 1290) in gradients of water and acetonitrile (Millipore Sigma). LC–MS data were collected using an Agilent (1260 Infinity II HPLC equipped with an InfinityLab LC/MSD mass detector). NMR data were collected using a 400 MHz Bruker Ascend. Compounds were dissolved in DMSO before testing, which was also used as vehicle controls throughout the assessment of compound activity.

## Crystallography

A single colorless blade crystal of **BP79** recrystallized from a mixture of methanol and DCM by slow evaporation. A suitable

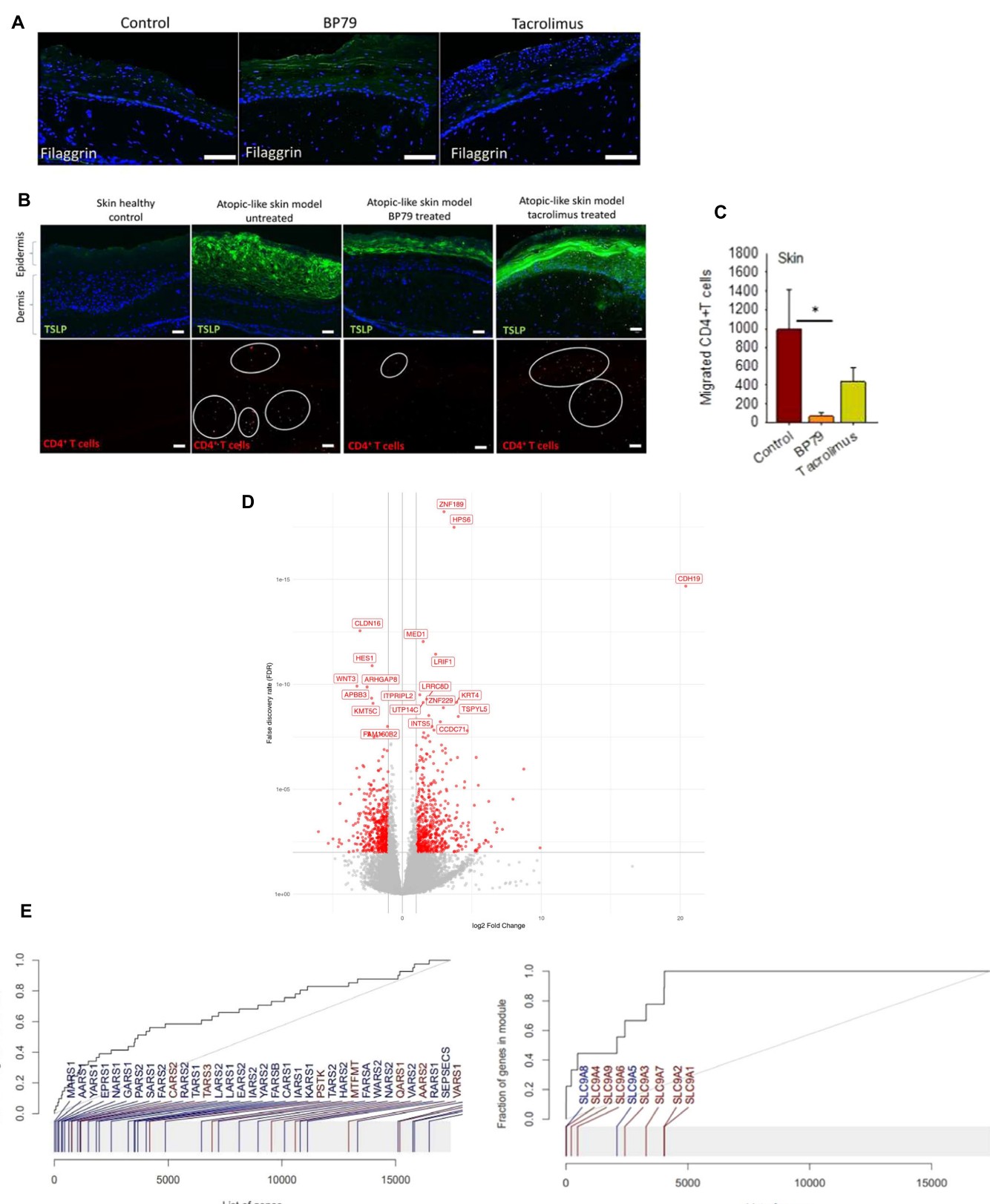

KEGG AMINOACYL tRNA biosynthesis pathway:
Lungs: treated versus untreated; adj. p-values 0.00013

REACTOME SODIUM PROTON EXCHANGERS pathway:
Lungs: treated  versus untreated; adj p -value 0.00048

**Figure 8. BP79 effects on atopic-like skin and lung models in an atopic diseases-organ chip.**

(A) Filaggrin expression in untreated (control), BP79-treated, and tacrolimus-treated atopic-like skin disease models. Scale bar = 50 µm. (B) TSLP expression and CD4 + T-cell infiltration in healthy, untreated and BP79 & tacrolimus-treated atopic-like skin disease models. BP79 treatment significantly reduced TSLP expression and CD4 + T-cell migration compared to untreated disease models. Scale bar = 50 µm. (C) Number of migrated CD4 + T cells were counted using ImageJ ($n = 5$). Data information: All data are presented as mean ± SEM. $n$ represents the number of biological replicates. Statistical analysis was performed using Student's $t$ test, *$P \leq 0.05$, **$P \leq 0.01$, ***$P \leq 0.001$. (D) Volcano plot showing the differential expression between treated and untreated diseased skin. Top 20 gene names are indicated on the plot. Genes significant at FDR < 0.01 with at least twofold difference between conditions are indicated by color. (E) Selected results of gene set enrichments in the contrast treated vs untreated diseased skin, co-cultured with lung tissue. Gene set enrichment evidence plots showing the gene set enrichment Receiving-Operator Characteristic (ROC) curve. Horizontal axis, genes ordered by $P$ value (genes in the gene set are highlighted). Vertical axis, fraction of the gene set covered. Statistical significance was obtained using the CERNO gene set enrichment test implemented in the R package tmod. aminoacyl tRNA synthesis (KEGG pathways gene set, IDhsa00970); sodium proton exchangers (REACTOME pathways gene set, ID R-HSA-425986). All gene set enrichments are significant at FDR < 0.001.

crystal with dimensions $0.12 \times 0.09 \times 0.03$ mm³ was selected and mounted on a mylar loop in oil on a Bruker APEX II area detector diffractometer. The crystal was kept at a steady $T = 115(2)$ K during data collection. The structure was solved with the **XT** 2018/2 (Sheldrick, 2015) solution program using Intrinsic Phasing methods and by using Olex2 (Dolomanov et al, 2009) as the graphical interface. The model was refined with **XL** (Sheldrick, 2015) using full matrix least squares minimization on $F^2$

***Crystal data***

$C_{10}H_8Cl_2N_2O_2$, $M_r = 259.08$, monoclinic, $P2_1/c$ (No. 14), a = 3.69390(10) Å, b = 22.9196(7) Å, c = 12.1562(4) Å, b = 93.042(2)°, a = g = 90°, $V = 1027.73(5)$ Å³, $T = 115(2)$ K, $Z = 4$, Z' = 1, $m(MoK_a) = 0.615$, 11708 reflections measured, 3073 unique ($R_{int} = 0.0359$) which were used in all calculations. The final $wR_2$ was 0.0969 (all data) and $R_1$ was 0.0404 ($I \geq 2 s(I)$).

| Compound | BP79 |
| --- | --- |
| Formula | $C_{10}H_8Cl_2N_2O_2$ |
| $D_{calc}$/g cm⁻³ | 1.674 |
| µ/mm⁻¹ | 0.615 |
| Formula weight | 259.08 |
| Color | Colorless |
| Shape | Blade |
| Size/mm³ | 0.12 × 0.09 × 0.03 |
| $T$/K | 115 (2) |
| Crystal system | Monoclinic |
| Space group | $P2_1/c$ |
| $a$/Å | 3.69390 (10) |
| $b$/Å | 22.9196 (7) |
| $c$/Å | 12.1562 (4) |
| $\alpha$/° | 90 |
| $\beta$/° | 93.042 (2) |
| $\gamma$/° | 90 |
| $V$/Å³ | 1027.73 (5) |
| $Z$ | 4 |
| $Z'$ | 1 |
| Wavelength/Å | 0.71073 |
| Radiation type | $MoK_\alpha$ |
| $\Theta_{min}$/° | 1.777 |

| Compound | BP79 |
| --- | --- |
| $\Theta_{max}$/° | 30.523 |
| Measured Refl's. | 11708 |
| Indep't Refl's | 3073 |
| Refl's I ≥ 2 σ(I) | 2570 |
| $R_{int}$ | 0.0359 |
| Parameters | 157 |
| Restraints | 0 |
| Largest Peak | 0.640 |
| Deepest Hole | −0.274 |
| GooF | 1.052 |
| $wR_2$ (all data) | 0.0969 |
| $wR_2$ | 0.0920 |
| $R_1$ (all data) | 0.0508 |
| $R_1$ | 0.0404 |

## Enzyme-linked immunosorbent assay (ELISA)

HuT78 and primary CD4 + T cells ($0.5 \times 10^6$ cells/ml) were activated with 100 ng/mL PMA, 1 µg/mL ionomycin, and 50 ng/mL TSLP for 36 h. TSLP inhibitors were added in selected wells. After 36 h, the culture media were collected, and the levels of IL-4 and IL-13 were determined using human IL-4 and IL-13 ELISA kits according to the manufacturer's instructions.

## MTT assay

HuT78 ($0.5 \times 10^6$ cells/ml), human primary CD4 + T ($0.5 \times 10^6$ cells/ml), keratinocytes ($0.5 \times 10^5$ cells/well) and fibroblast ($0.5 \times 10^5$ cells/well) cells were treated with a range of different TSLP inhibitor concentrations for 24 h. Subsequently, MTT (5 mg/ml) reagent was added for 4 h. Acidified isopropanol or dimethyl sulfoxide (DMSO) were added to solubilize the formazan crystals, and the absorbance was measured at 570 nm using a microplate reader (BioTek Instruments, USA). Cell viability was determined as a percentage of the vehicle control group treated with a medium containing 0.2% DMSO.

## Immunoblotting

Western blot analysis was performed in HuT78, primary CD4⁺ T cells, and keratinocytes treated with TSLP inhibitors to determine

the target binding and inhibition of TSLP-mediated downstream signaling pathway. Following treatment with TSLP for 20 min in CD4 + T cells and HuT78 cells and for 24 h in keratinocytes, the cells were lysed with RIPA buffer containing a protease and phosphatase inhibitor cocktail. Total protein concentration was determined with a BCA assay kit. Twelve micrograms of total protein were separated on sodium dodecyl sulfate-polyacrylamide gel electrophoresis (SDS-PAGE) gels and then transferred onto polyvinylidene fluoride (PVDF) membranes. The levels of indicated proteins were blotted by incubating with primary antibodies (1:1,000) overnight at 4 °C, followed by incubation with secondary antibodies (1:5000) for 1 h at room temperature (RT). Immunoreactive proteins were visualized using an Infrared western blot imaging system (Odessy CLx).

## Dendritic cell activation and co-culture with CD4 + T cells

PBMCs were isolated from buffy coats of healthy donors using Lymphoprep. Human myeloid dendritic cells (mDCs) were subsequently isolated from the PBMCs using the EasySep™ human myeloid DC enrichment kit following the manufacturer's protocol. The dendritic cells were cultured in human dendritic cell culture media overnight. Cells were seeded at a density of $2 \times 10^5$ cells/200 μl medium in round-bottomed 96-well plates. The next day, the cells were stimulated with human recombinant TSLP at a concentration of 50 ng/ml for 24 h. Selected wells were treated with 20 μM of BP79, along with an equal volume of DMSO. Following the treatments, cells were stained with a PE-conjugated OX-40L antibody and analyzed using a CytoFLEX flow cytometer. The production of CCL17 in the culture supernatants after 24 h was determined by ELISA using R&D Systems kits.

To induce dendritic cell-mediated Th2 cell differentiation, mDCs were co-cultured with naive CD4 + T cells in 96-well round-bottom culture plates. First, mDCs ($2 \times 10^4$/well) were incubated with 50 ng/ml TSLP for 24 h. Selected wells were treated with 20 μM of BP79 or an equal volume of DMSO. After 24 h, naive CD4 + T cells ($1 \times 10^5$/well) were added and co-cultured for 7 days. Ad day 7, the cells were re-stimulated with 100 ng/mL PMA and 1 μg/mL ionomycin for another 24 h. Cell culture supernatants were collected, and IL-13 expression was detected using ELISA.

## Proximity ligation assay (PLA)

Proximity ligation assay was performed to determine the protein-inhibitor binding efficacy. Primary keratinocytes were cultured in coverslips overnight. Next day, cells were treated with 10 μM TSLP inhibitors and equal volume of DMSO as a vehicle control and incubated for 1 h at 37 °C, 5% $CO_2$. Cells were then stimulated with 50 ng/ml TSLP for 30 mins and subsequently fixed in 4% formaldehyde. Samples were processed using a DuoLink PLA kit with Anti-Mouse PLUS and Anti-Goat MINUS, and Red PLA detection reagent, and primary antibodies for TSLPR and IL-7Rα, according to the manufacturer's protocol. Red fluorescence was detected using the Texas Red filter. In total, 25 to 40 images were captured per group using EVOS M5000 fluorescence microscope (Thermo Scientific, USA). Data analysis was performed using ImageJ software according to a previously published protocol (Gomes et al, 2016).

## In vitro thermal shift assay (TSA)

The assay was performed as described previously [10]. Briefly, recombinant TSLPR protein was dissolved in 25 mM HEPES (pH = 7.4), 150 mM NaCl and 1 mM DTT at a final concentration of 0.5 μg/ml. TSLPR protein was incubated with 20 μM BP79, along with DMSO control, in 50 μL reaction buffer at 25 °C for 10 min, and then heated at various temperatures for 3 min. Samples were centrifuged at 14,000 rpm for 10 min at 4 °C. In all, 30 μl of supernatants were collected carefully from the top and mix with 4x Laemmli sample buffer. TSLPR protein levels were measured by western blotting.

## Kinase panel screening

BP79 was tested at 10 μM in the KinomeScan (97 kinases) radioligand displacement assay from Eurofins Discovery Services in duplicate. Values displayed in Fig. 4C are the average of the duplicates from a single experiment.

## Generation of the 3D epithelial models

A skin disease model that closely emulates atopic dermatitis in vitro was prepared according to previously established procedures (1). In brief, the filaggrin gene (FLG) was knocked down in keratinocytes using a siRNA (Invitrogen, Cat# 87663809) with HiPerFect transfection reagent (Qiagen, Cat# 301705). After 24 h, dermal Fbs ($2.1 \times 10^4$ cells/well), FBS (Sigma, F1051) (21 μL/well) and type I bovine collagen (Advanced BioMatrix, Cat# 5005) were brought to neutral pH and poured into transwell insert for 24-well plates (Corning, Cat# 353504) with a growth area of 0.33 cm² (Costar, Cat# 3422). After incubation for 1 h at room temperature and another hour at 37 °C, 200 μL of EpiLife culture medium was added, and the system was transferred to a 37 °C incubator with 5% (v/v) CO2 and 95% (v/v) humidity for another 2 h. Subsequently, $3 \times 10^5$ cells/insert normal and FLG knockdown KCs (were added on top of the collagen matrix. After 24 h, the model was lifted to the air-liquid interface and the medium was changed to the keratinocyte differentiation medium (KDM). The skin models were cultured for 11 days (at 37 °C, with 5% CO2 and 95% humidity). Medium change was performed every other day.

Bronchial epithelial models were generated as described elsewhere (2). Briefly, normal human lung Fibroblasts ($1.9 \times 10^4$ cells/well) and FBS (Sigma, F1051) (9.5 μL/well) were embedded in type I bovine collagen (Advanced BioMatrix, Cat# 5005) mimicking the bronchial epithelial stroma. The mixture was brought to neutral pH and decanted (total volume per well is 79.2 μL) into 3D transwell permeable supports insert for 24-well plates (Corning, Cat# 353504) with a growth area of 0.33 cm² (Costar, Cat# 3422). After incubation for 2 h at 37 °C, 200 μL of PneumaCult-Ex medium (STEMCELL, Cat# 05008) were added and the system was transferred to a 37 °C incubator with 5% (v/v) $CO_2$ and 95% (v/v) humidity for another 2 h. After this incubation, NHBEs in PneumaCult-Ex medium (STEMCELL, Cat# 05008) ($2.65 \times 10^5$ cells/insert) were added on top of the collagen matrix. After 24 h, the model was lifted to the air-liquid interface and the medium was changed to PneumaCult–ALI medium (STEMCELL, Cat# 05001). The bronchial epithelial models were cultured for 18 days (at 37 °C, with 5% $CO_2$ and 95% humidity). Medium change was performed every other day.

## Skin permeation testing

Skin permeation of BP79 was evaluated using the Franz cell setup. The acceptor compartments of the Franz cells were filled with phosphate-buffered saline (PBS; pH 7.4, 32 °C). Punched disks of excised human skin (CREB approval # H19-03096) were mounted onto the Franz diffusion cells (static type, volume 12 mL, diameter 15 mm), with the stratum corneum exposed to air and the dermis in contact with the PBS. Following 30 min equilibration time, 100 µL of 20 µM BP79 or vehicle control was applied topically on the skin. The Franz cells were sealed with Parafilm® to prevent evaporation of the test solution. To determine the amount of permeated BP79, buffer solution from the acceptor medium was sampled after 2, 4, 8, and 24 h, respectively and stored at −20 °C. For LC/MS analysis, the samples were extracted twice using 10 mL of ethyl acetate. The extracts were combined and evaporated to dryness under nitrogen at 45 °C and reconstituted in 100 µL acetonitrile. Subsequently, BP79 detection was performed using the UHPLC/MS/MS system consisted of an Agilent 1290 Infinity Binary Pump, a 1290 Infinity Sampler, a 1290 Infinity Thermostat, and a 1290 Infinity Thermostatted Column Compartment (Agilent, Mississauga, Ontario, Canada) connected to an AB SCIEX QTRAP® 5500 hybrid linear ion trap triple quadrupole mass spectrometer equipped with a Turbo Spray source (AB SCIEX, Concord, Ontario, Canada).

Chromatographic separation was performed using an InfinityLab Poroshell 120 EC-C18, $3 \times 50$ mm, 1.9 µm particle size (Agilent, Santa Clara, CA, USA) with column temperature of 30 °C. The mobile phase consisted of 0.1% formic acid in deionized water as solvent A and 0.1% formic acid in acetonitrile as solvent B with the flow rate of 450 µL/min. The LC gradient was started with 90% of solvent A at initial condition ($t = 0$ min) and decrease to 10% in 4 min ($t = 4$ min), then held for 1.5 min ($t = 5.5$ min). The gradient return to the initial condition of 90% solvent A and equilibrate for 1.4 min ($t = 7$ min) before the next injection. The injection volume was 5 µL. The mass spectrometer was operated in positive ionization mode with the following conditions: Curtain gas 30 units, Collision gas (CAD) high, Ionspray 5500 V, temperature 450 °C, ion source gas 1, 40 units, ion source gas 2, 60 units. Nitrogen gas was used for curtain gas, collision gas, ion source gas 2 (vaporizing gas), and zero air was used for ion source gas 1 (nebulizing gas). Declustering potential 100, Entrance potential 10, Resolution Q1 Unit, Resolution Q3 Unit, and dwell time was 150 msec. The MRM transitions were $m/z$ $259.0 \rightarrow 214.0$ with collision energy (CE) 21, collision cell exit potential (CXP) 4 and $m/z$ $259.0 \rightarrow 242.1.0$ with CE 17, CXP 6. Data were acquired using the Analyst 1.5.2. software on a Microsoft Windows XP Professional operating platform.

## Generation of 3D immunocompetent skin models supplemented with activated CD4 + T cells

Normal (FLG + ) and filaggrin-deficient (FLG − ) skin models were generated using the methods described above. Primary human CD4 + T cells were activated using ImmunoCult Human CD3/CD28 T Cell Activator for 24 h. At day 12, $0.3 \times 10^6$ activated CD4 + T cells were applied underneath the dermis directly onto the cell culture insert membrane of the skin models and cultured for 2 more days. Skin models were treated with 20 µM TSLP inhibitor-

BP79 three times topically, models treated with vehicle control (DMSO) and medium only served as control. After end of the treatment, culture media were collected for cytokine array assay.

## Cytokine array assay

Culture media were collected from the 3D immunocompetent skin tissue models. Cytokines including TSLP, IL-2, IL-4, IL-5, IL-9, IL-10, IL-13, IL-22, TNF-α, and IFN-γ, levels were quantified by a multiplex assay U-PLEX platform with electrochemiluminescent enzyme-linked immunosorbent assays [Meso Scale Discovery (MSD), USA]. The assay was performed according to the manufacturer's instructions, and data were acquired by MESO QuickPlex SQ 120 plate reader.

## Organ-on-chip-based co-cultivation of skin (disease) and bronchial epithelial model

HUMIMIC Chip3plus were purchased from TissUse (Germany). Prior to chip connection, the HUMIMIC starter control unit (TissUse, Berlin, Germany) was configured to 30 bpm (air flow of 1.5 L/min) with +500 mbar pressure-out and -500 mbar vacuum-out, mimicking recirculation mode of the capillaries (3). With this setup, HUMIMIC Chip3plus was first placed in a 37 °C incubator for 24 h. Subsequently, the chips were connected to the HUMIMIC starter and were perfused for 24 h. Lastly, the media in the chip were changed into an 800 µL medium mix (1:1:1 of KDM, PneumaCult–ALI medium, and complete RPMI medium, hydrocortisone was depleted from KDM and PneumaCult–ALI medium as glucocorticoids suppress T-cell function (4)). The chip was then perfused for 24 h. After the preparation, $5 \times 10^5$ activated CD4 + T cells in fresh 800 µL medium mix were applied to the chip. The FLG knockdown skin model and bronchial epithelial model were transferred into the chip, avoiding the formation of air bubbles and start perfusion. The day of transfer is considered day 1 of the OOC cultivation. In all, 10 µL of a 20 µM solution of the anti-TSLP compounds (BP79) were topically applied onto the skin model on day 1–4 h since the starting of perfusion. On day 2, the OOC system was replenished with fresh $5 \times 10^5$ activated CD4 + T cells in 800 µL medium mix. BP79 was applied to the skin model after four hours of replenishment. After 24 h (on day 3), BP79 was applied to the skin model again. After another 24 h (day 4), final BP79 application was performed and both skin and bronchial epithelial models were harvested in 4 h. Here, the culture medium was collected, the skin models were cry-frozen and bronchial epithelial models were paraffin-embedded for further characterizations.

## H&E staining

Sections of skin models (6 µm) and bronchial epithelial models (4 µm) were obtained using a cryotome or microtome for histological analyses. Hematoxylin (Thermo Scientific, Cat# 6765007) and eosin (Thermo Scientific, Cat# 6766007) (H&E) staining was performed according to standard staining procedures.

## Immunofluorescence assay

Tissue sections were fixed using 4% formaldehyde for 10 min at room temperature. Then, tissues were permeabilized with 0.5% TritonX-100 in PBS for 10 min at room temperature; washed with

**Table 1. Primer sequences for qPCR**

| Gene | Primer sense 5′-3′ | Primer antisense 5′-3′ |
| --- | --- | --- |
| GAPDH | CTCTCTGCTCCTCCTGTTCGAC | TGAGCGATGTGGCTCGGCT |
| FLG | AAGGAACTTCTGGAAAAGGAATTTC | TTGTGGTCTATATCCAAGTGATCCAT |
| TSLP | CCCAGGCTATTCGGAAACTCAG | CGCCACAATCCTTGTAATTGTG |
| KRT5 | AGTTTGTGATGCTGAAGAAG | GTTAATCTCATCCATCAGTGC |
| KRT10 | AAGAGCAAGGAACTGACTAC | CGTCTCAATTCAGTAATCTCAG |
| KRT14 | AGATCAAAGACTACAGTCCC | ACTCTGTCTCATACTTGGTG |

PBS containing 0.0025% BSA and 0.025% Tween 20 and blocked with normal goat serum (1:20 in PBS). The sections were incubated overnight at 4 °C with primary antibodies (in PBS, 0.0025% BSA, 0.025% Tween 20). Subsequently, the sections were incubated for an additional 1 h at room temperature with secondary antibodies (1:400 in PBS, 0.0025% BSA, 0.025% Tween 20) and counterstained with DAPI mounting medium. Slides were then imaged using EVOS M5000 fluorescence microscope.

### LDH and glucose assay

Lactate dehydrogenase (LDH) and glucose concentrations of the medium in the chip were monitored. Here, medium was collected at day 2 and day 4 and then the LDH concentration was determined using the cytotoxicity detection kit PLUS (Roche, Cat# 04744934001) following the manufacturer's instructions. Glucose consumption in the OOC system was measured using the glucose LiquiColor® (Stanbio, Cat# 1070-125) following the manufacturer's instructions.

### Real-time quantitative polymerase chain reaction (qPCR)

Primer sequences are listed in Table 1. For gene expression analysis, total RNA was extracted from the skin and lung models. Briefly, models were lysed in PureLink RNA Mini Kit lysis buffer and then homogenized for 30 s at 25 Hz using a TissueLyzer (Qiagen, Germany). RNA was isolated according to the manufacturer's protocol. Total RNA was quantified using Nanodrop 2000 (Thermo Scientific, USA). cDNA was synthesized, using iScriptTM cDNA Synthesis Kit. qPCR was performed with SsoAdvanced Universal SYBR Green Supermix using a StepOnePlus™ Real-Time PCR System (Applied Biosystems, USA). The following Primer sequences were used. Glyceraldehyde-3-phosphate dehydrogenase (GAPDH) served as a housekeeping gene.

### RNA-Seq processing and analysis

The reads were aligned to the human genome, v. GRCh38, p7 using the STAR aligner, v. 2.7.8.a (Dobin et al, 2013). For quality control, we used FastQC and MultiQC (Ewels et al, 2016). Read counts were collected using featureCounts (Liao et al, 2013). Features with fewer than a total of ten counts or fewer than five samples with at least three counts were eliminated.

For differential gene expression analysis, we used the R package DESeq2 (Love et al, 2014), v. 1.30.1. Gene set enrichment testing was done with the CERNO algorithm implemented in the tmod

**The paper explained**

**Problem**
Thymic stromal lymphopoietin (TSLP) is a pro-inflammatory cytokine that drives many inflammatory diseases including eczema, which has sparked great interest for therapeutic targeting. However, so far very few drugs are available and all of them are proteins which are expensive and must be systemically administered.

**Results**
We describe the development of the first putative, non-proteinic TSLP inhibitors which can also be applied locally onto the skin, thus significantly expand the treatment options for eczema patients. We used a combined approach of structure-based virtual screening and docking of >1,000,000 compounds, chemical optimization, and iterative biological testing focused on translational methodology. This led to the identification of our lead compound BP79 which effectively abrogates TSLP-mediated effects and significantly downregulates eczema-relevant pro-inflammatory pathways and cytokines.

**Impact**
We report the first potent small-molecule TSLPR inhibitor which has the potential to expand the therapeutic options in eczema and beyond.

package (Zyla et al, 2019), v. 0.50.7. with gene sets sourced from the MSigDB using the msigdbr package, v. 7.4.1. P values were corrected for multiple testing using the Benjamini–Hochberg method (Benjamini and Hochberg, 1995).

### Statistical analysis

For each experiment, at least three biological replicates (cells/tissue from ≥ three different donors; no sample blinding or randomization) were performed; the data are shown as the mean ± standard error. Data were analyzed using Sigma Plot version 10.0. Unpaired or paired Student's $t$ test was used with $P \le 0.05$ considered statistically significant. The exact $P$ values indicated in the manuscript figures are listed in Appendix Table 10.

## Data availability

Primary transcriptomic datasets produced in this study are deposited in a public database, and can be accessed through this link: https://www.ncbi.nlm.nih.gov/geo/query/acc.cgi?acc=GSE264457.

The source data of this paper are collected in the following database record: biostudies:S-SCDT-10_1038-S44321-024-00085-3.

## Peer review information

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

## Acknowledgements

The authors would like to thank Dr. Katharina Hoerst (The Freie University of Berlin, Berlin, Germany) for initiating and optimizing the assay conditions using Hut78 cells, Dickson Lai (UBC Faculty of Pharmaceutical Sciences, Canada) for

his assistance with the skin permeation assay sample preparation, LC–MS detection and data analysis, Andy Kim (UBC Faculty of Pharmaceutical Sciences, Canada) for assistance with the qPCR detection of filaggrin expression from primary keratinocytes cells. We would also like to thank the staff members from the Sequencing Core Facility (Biomedical Research Centre, UBC, Vancouver, Canada) for performing RNA sequencing and TissUse (Berlin, Germany) for measuring LDH and glucose concentration from the organ-on-a-chip samples. The crystal structure of BP79 was determined by Dr. Brian O Patrick, UBC Department of Chemistry, Structural Chemistry Facility (Appendix Tables S2–S7). The authors greatly acknowledge financial support from the Canadian Institutes for Health Research (CIHR; AWD-016113 CIHR 2020), the Faculty of Pharmaceutical Sciences at UBC (SH and BDGP), the BC Lung Foundation (SH) and the Foundation Charité (SH). This project was further supported by grants from NSERC CGS-M (CH), BDGP is a Michael Smith Health Research BC Scholar (18292). PPA was supported by the Canadian Allergy, Asthma, and Immunology Foundation (CAAIF) and Sanofi Genzyme Canada. The funders had no role in study design, data collection and analysis, decision to publish, or preparation of the manuscript.

## Author contributions

**Partho Protim Adhikary**: Conceptualization; Data curation; Investigation; Visualization; Methodology; Writing—original draft; Writing—review and editing. **Temi Idowu**: Investigation; Visualization; Methodology; Writing—review and editing. **Zheng Tan**: Investigation; Visualization; Methodology. **Christopher Hoang**: Investigation; Methodology. **Selina Shanta**: Investigation; Methodology. **Malti Dumbani**: Investigation; Methodology; Writing—original draft. **Leah Mappalakayil**: Investigation; Visualization. **Bhuwan Awasthi**: Investigation; Methodology. **Marcel Bermudez**: Supervision; Methodology. **January Weiner**: Conceptualization; Software; Formal analysis; Visualization; Writing—original draft. **Dieter Beule**: Resources; Software; Supervision. **Gerhard Wolber**: Resources; Supervision; Visualization; Methodology; Writing—original draft; Project administration. **Brent DG Page**: Conceptualization; Data curation; Supervision; Funding acquisition; Visualization; Methodology; Writing—original draft; Project administration; Writing—review and editing. **Sarah Hedtrich**: Conceptualization; Resources; Data curation; Formal analysis; Supervision; Funding acquisition; Validation; Visualization; Methodology; Writing—original draft; Project administration; Writing—review and editing.

Source data underlying figure panels in this paper may have individual authorship assigned. Where available, figure panel/source data authorship is listed in the following database record: biostudies:S-SCDT-10_1038-S44321-024-00085-3.

## Funding

## Disclosure and competing interests statement

The authors declare no competing interests. A patent application has been filed for the discovery of BP79. No further competing interests are being declared.

# Expanded View Figures

**Figure EV1.   Biological screen of twenty-eight BP79 analogs.**

(A) Chemical structures of BP79 analogs produced. (B) IL-13 expression in HuT78 cells after treatment with 1μm of inhibitor for 36 h ($n = 2$). (C) IL-4 expression in HuT78 cells after treatment with 1 μm of inhibitor for 36 h. Bars represent mean percentage cytokine expression relative to the DMSO control ($n = 2$). Non-treated (NT), PMA-ionomycin stimulated (PI), PMA-ionomycin and TSLP-stimulated (PI + TSLP), and a PMA-ionomycin + TSLP-stimulated 0.4% DMSO vehicle control (DMSO) were included. Data information: All data are presented as mean ± SEM. $n$ represents the number of biological replicates. Statistical analysis was performed using Student's $t$ test, $*P \leq 0.05$, $**P \leq 0.01$, $***P \leq 0.001$.

▶

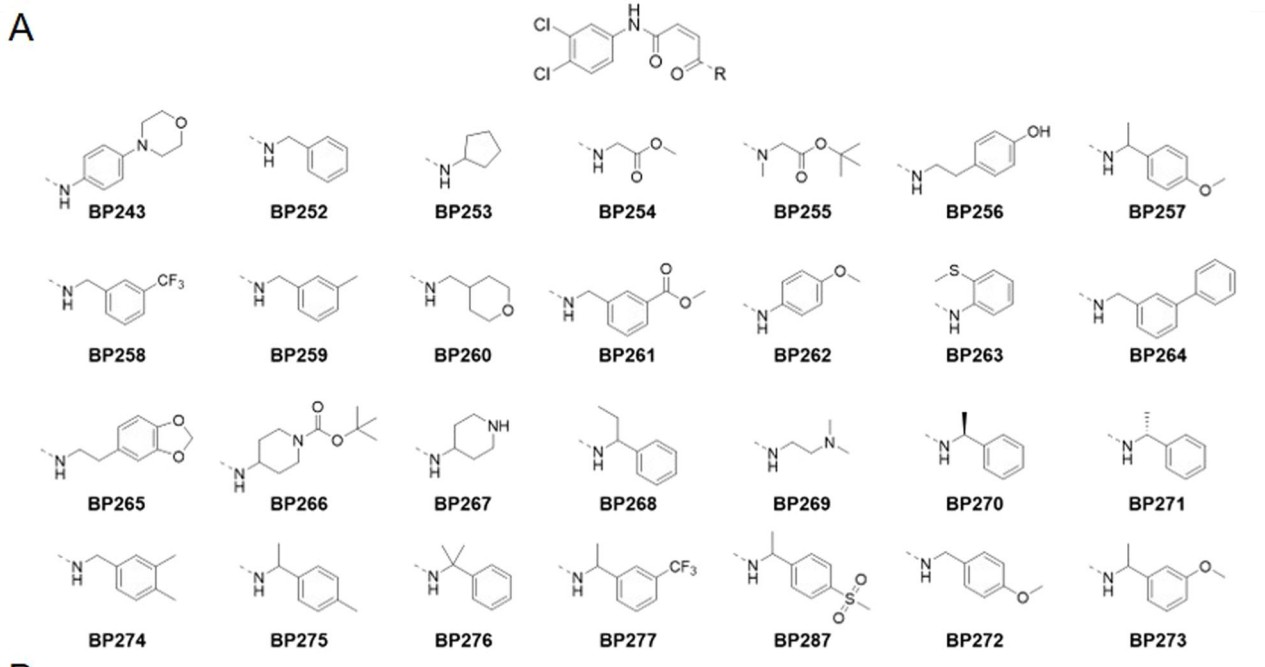

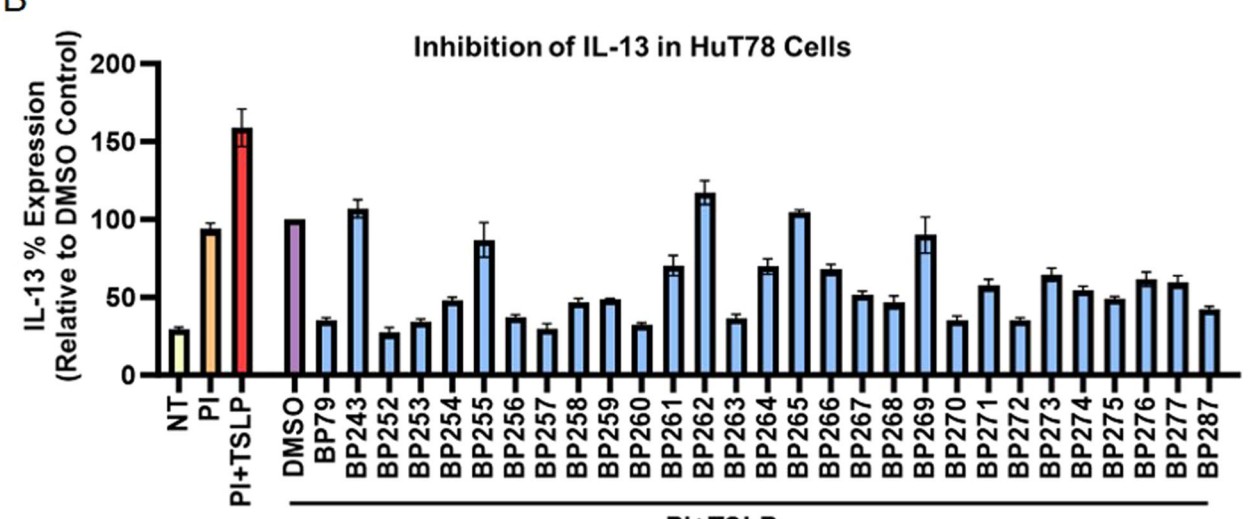

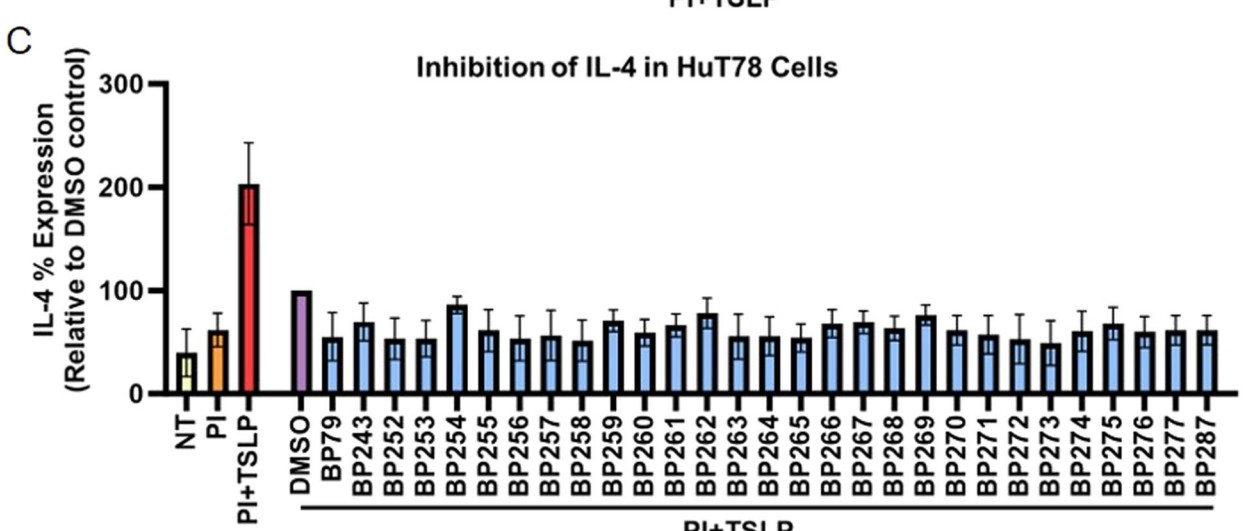

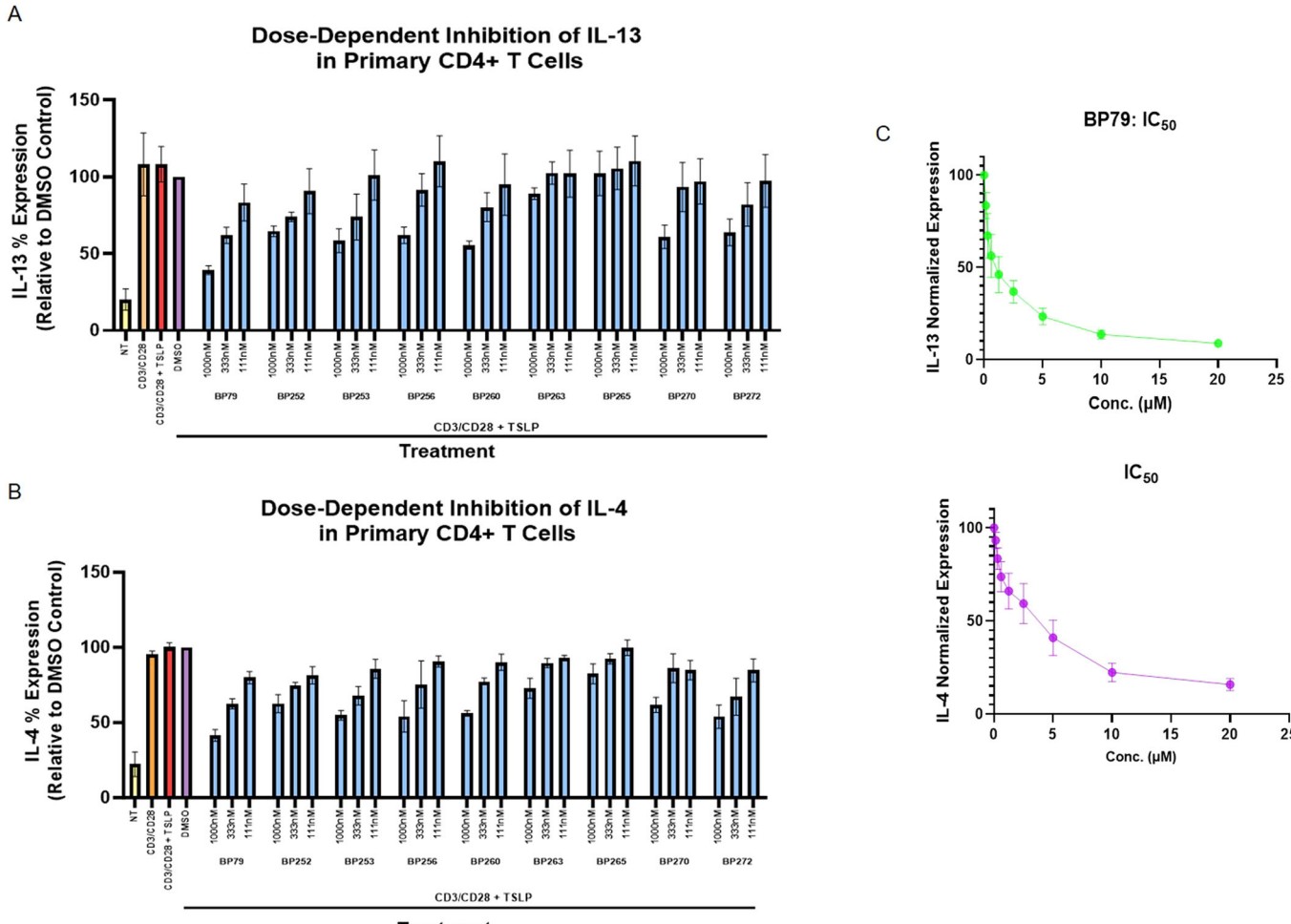

**Figure EV2. Dose-dependent screen of BP79 analogs in CD4 + T cells.**

(A) IL-13 expression in primary CD4 + T cells after treatment with 111 nM, 333 nM, and 1000 nM of inhibitor for 36 h. (B) IL-4 expression primary CD4 + T cells after treatment with 111 nM, 333 nM, and 1000 nM of inhibitor for 36 h. Bars represent mean percent cytokine expression relative to the DMSO control, $n = 3$. Non-treated (NT), CD3/CD28 stimulated (CD3/CD28), CD3/CD28 and TSLP-stimulated (CD3/CD28 + TSLP), and vehicle (CD3/CD28 + TSLP + DMSO) controls were included. (C) Dose–response curves of BP79, which were used to calculate the $IC_{50}$ value and HillSlope ($n = 3$). Data information: All data are presented as mean ± SEM. $n$ represents the number of biological replicates. Statistical analysis was performed using Student's $t$ test, $*P \leq 0.05$, $**P \leq 0.01$, $***P \leq 0.001$.

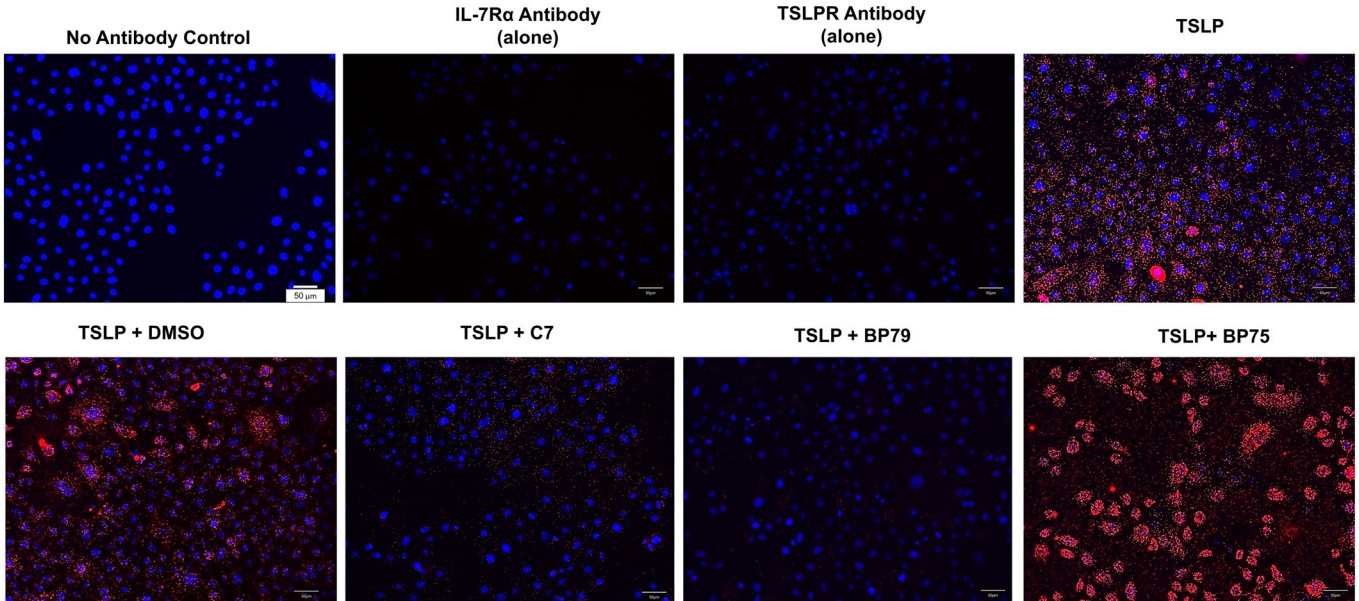

**Figure EV3.   Full panel images of proximity ligation assay (PLA).**

PLA was performed to check the inhibition of TSLP-mediated ternary complex formation with BP79. Primary keratinocyte cells were treated with the TSLP inhibitors and stimulated with TSLP. Cells were fixed and the interaction between TSLPR and IL-7Rα was observed with PLA assay. C7 and BP79 inhibited the TSLP:TSLPR:IL-7Rα ternary complex formation. Scale bar = 50 μm.

