## [Peer Review File · EMBO Molecular Medicine]

Disrupting TSLP-TSLP receptor interaction via putative small molecule inhibitors

Sarah Hedtrich, Partho Adhikary, Temi Idowu, Zheng Tan, Christopher Hoang, Selina Shanta, Malti Dumbani, Leah Mappalakayil, Bhuwan Awasthi, Marcel Bermudez, January Weiner, Dieter Beule, Gerhard Wolber, and Brent Page

Corresponding authors: Sarah Hedtrich (sarah.hedtrich@bih-charite.de) , Brent Page (brent.page@ubc.ca)

Review Timeline:

Submission Date:	9th Aug 23
Editorial Decision:	4th Sep 23
Appeal:	7th Sep 23
Editorial Decision:	29th Sep 23
Revision Received:	15th Apr 24
Editorial Decision:	25th Apr 24
Revision Received:	13th May 24
Editorial Decision:	17th May 24
Revision Received:	21st May 24
Accepted:	23rd May 24

Editor: Lise Roth

Transaction Report:

1st Sep 2023

Decision on your manuscript EMM-2023-18490

Dear Dr. Hedtrich,

Thank you for the submission of your manuscript to EMBO Molecular Medicine. We have now received feedback from the two referees who had agreed to review your manuscript.

As you will see from the reports below, while the referees acknowledge the potential translational interest of the work, they also mention several shortcomings that unfortunately make the study too preliminary at this stage for consideration at EMBO Molecular Medicine. We encourage a single round of revisions in a limited timeframe, and based on the referees' reports, I am afraid I see little choice but to return the manuscript to you at this point with the decision that we cannot offer to publish it.

While we cannot pursue this manuscript further, we encourage you to transfer your study to our not-for-profit open-access sister journal, Life Science Alliance (LSA). We shared your manuscript and the accompanying reviews with LSA Executive Editor, Eric Sawey, who is interested in these findings, and would like to invite further consideration of this manuscript at LSA, revised to address the Reviewers' comments.

We understand that such a revision might need to be re-reviewed, in which case, Dr. Sawey will walk the Reviewers through our transfer process.

We encourage you to use the link below to transfer your manuscript to LSA. You do not need to revise the manuscript before transferring it to LSA. Once you transfer, Dr. Sawey will email you an invitation to revise and resubmit, listing the same revision requests as mentioned above. Please feel free to reach out at e.sawey@life-science-alliance.org if you have any questions about the LSA journal, the transfer process or the revisions requested.

I am very sorry to disappoint you on this occasion and I hope you will view the possibility of a transfer favorably. If this is the case, please use the link below to transfer the manuscript directly.

With kind regards,

Lise Roth

Lise Roth
Senior Editor
EMBO Molecular Medicine

***** Reviewer's comments *****

Referee #1 (Comments on Novelty/Model System for Author):

The use of activated CD4 T cells and keratinocytes as targets for TSLP-mediated responses are not cell types generally used for this purpose.

Referee #1 (Remarks for Author):

The generation of a small molecule inhibitor of TSLP signaling would be very important and impactful. While the data presented are interesting but insufficient for the conclusions drawn in the manuscript. There are some issues that need to be addressed before publication can be considered:

1. Based on the author's description, BP79 (the leading candidate) should act as a binding inhibitor of TSLP to TSLPR. It is important that the authors demonstrate binding of TSLP to TSLP in the presence and absence of BP79.

2. The authors need to use ex vivo systems that more accurately reflect TSLP function for their analyses of BP79, etc. CD4 T cells generally do not express TSLPR until activated, and the role of TSLP in cytokine production is unclear. Keratinocytes are TSLP producers, and not generally known as responders. The authors need to examine the ability of these molecules to inhibit antigen-presenting cell-mediated Th2 differentiation and TSLP-mediated CCL17 production.

Referee #2 (Comments on Novelty/Model System for Author):

In this study, authors developed a small-molecule TSLP receptor (TSLPR) inhibitor called BP79. They evaluated its efficacy using human primary CD4+ T cells and a human atopic-like skin disease model (using microfluidic multi-organ chips). They showed that topical application of BP79 suppressed immune cell infiltration and production of IL-13, IL-4, TSLP, and periostin, while upregulating filaggrin, a protein related to the skin barrier. Since small molecule TSLP inhibitors, which would allow a topical application, have not yet been developed, their findings cover an essential topic for TSLP-related allergic disease research, such as atopic dermatitis and asthma. Although this paper has the potential to be published, the manuscript contains several issues of concern.

Referee #2 (Remarks for Author):

In this study, authors developed a small-molecule TSLP receptor (TSLPR) inhibitor called BP79. They evaluated its efficacy using human primary CD4+ T cells and a human atopic-like skin disease model (using microfluidic multi-organ chips). They showed that topical application of BP79 suppressed immune cell infiltration and production of IL-13, IL-4, TSLP, and periostin, while upregulating filaggrin, a protein related to the skin barrier. Whereas their findings cover an essential topic for TSLP-related allergic disease research, the manuscript contains several issues of concern.

Major points:

1. Whereas it has been well shown that BP79 suppresses Th2 cytokines such as IL-13 and IL-4, phenotypical evidence is not sufficient. In Fig.6-c, the skin histopathology of the atopic condition treated with BP79 still seems to have hyperkeratosis and acanthosis. The authors need to show more convincing quantitative data on the phenotypic improvement by BP79.
2. The authors claimed that BP79 suppressed immune cell skin infiltration in the abstract. However, they only showed immunohistochemistry of the human atopic-like skin disease model (AD-model) in Figure 6F, demonstrating that AD-model treated by BP79 has few CD4+ T cell skin infiltration, while the AD-model had rich CD4+ T cell skin infiltration. The authors need to show the quantitative data on these findings. In addition, these results may be due to suppression of CD4+ T cell proliferation or induction of apoptosis of CD4+ T cells by BP79. The authors should explain how BP79 suppressed immune cell infiltration into the skin.
3. Please discuss reasons why the efficacy of tezepelumab in atopic dermatitis is inadequate and limited. In contrast, in this study the authors demonstrated BP79 suppressed significantly AD inflammation. This discrepancy may give readers a little concern about whether BP79 indeed improves atopic dermatitis in clinical settings.
4. To prove that the transcriptional profiles of BP79-treated diseased skin are more similar to healthy skin than untreated diseased skin, the author compared the number of DEGs. However, the difference in transcriptional profiles should be interpreted not by the number of DEGs but by the contents of DEGs. For example, did the transcriptome of cytokines associated with atopic dermatitis change with or without treatment with BP79? Alternatively, clustering, such as PCA, might allow for qualitative evaluation if healthy skin and treated skin were included in the same cluster, separate from untreated skin.
5. Relating to the comment above, the interpretation of RNA-seq results requires some caution. The results of RNA-seq demonstrated the enriched KEGG-Ribosome pathway as Fig.S8 showed many genes of ribosomal proteins. However, ribosomal RNA is not used for general RNA-seq analysis and a high quantity of rRNA could affect the quality of the sample or analysis. If rRNA is removed during the analysis phase, different pathways may be found.

Minor points:

1. Y axis of Figure 6D is written as "% of expression." It is unclear whether this was the expression of mRNA or protein. If this is mRNA expression, the immunohistochemistry of indicated treatments in Figure 6D will add stronger evidence of the efficacy of BP79.
2. Can the authors quantify the CD4+ T cell density in Figure 6F?
3. Line 170, is "IL-7R α " a mistake for "IL-7Ra"?
4. In the legend for Fig3, there are two explanation sentences for Fig.3-f, the former seems to be inappropriate.

As a service to authors, EMBO provides authors with the possibility to transfer a manuscript that one journal cannot offer to publish to another EMBO publication. The full manuscript and if applicable, reviewers reports are automatically sent to the

receiving journal to allow for fast handling and a prompt decision on your manuscript. For more details of this service, and to transfer your manuscript to another EMBO title please click on Link Not Available

06.09.2023

Dear Mrs. Roth,

First of all, we would like to thank you for handling our submission. After reviewing the comments carefully, we, however, feel the urgent need to address particularly the comments of Reviewer 1 since we have major concerns about their validity. Further, it appears that they may have not fully read the manuscript as many of the concerns/comments are in fact addressed in the manuscript.

Therefore, please allow us to address Reviewer 1 comments in more detail as it appears that they, though not valid, significantly contributed to the decision 'reject'.

1. Cell Types Used for TSLP-Mediated Responses:

Reviewer: The use of activated CD4 T cells and keratinocytes as targets for TSLP-mediated responses are not cell types generally used for this purpose.

Our Response: We respectfully but strongly disagree with this statement. We deliberately selected keratinocytes and activated CD4+ T cells as primary cell types for specific reasons. The objective of our project was to develop a small molecule TSLP inhibitor for the treatment of atopic dermatitis (AD), a Th2-dominated condition in which TSLP is a central player.

In AD, skin barrier dysfunction occurs due to factors such as filaggrin gene mutations, microbial infections, or mechanical disruption. In this context, keratinocytes are the main source of TSLP, which triggers further inflammation.

TSLP secreted by keratinocytes activates CD4+ T cells, either directly or through dendritic cells. These activated CD4+ T cells subsequently produce Th2 cytokines, including IL-4, IL-5, IL-9, and IL-13. Although other immune cells can also produce Th2 cytokines, CD4+ T cells are the primary source. Moreover, TSLP plays a crucial role in differentiating CD4+ T cells into pathogenic Th2 cells. In summary, keratinocytes produce TSLP and activate CD4+ T cells, which, in turn, produce Th2 cytokines. There is a large body of evidence corroborating these claims [1-10] and, thus, fully justifying the use of these cell types for drug discovery in AD.

Further, the reviewer does not provide any suggestions for alternative cell types and, honestly, we cannot think of cell types that may be better suited for our purpose.

2. Binding Inhibition of TSLP to TSLPR:

Reviewer: Based on the author's description, BP79 (the leading candidate) should act as a binding inhibitor of TSLP to TSLPR. It is important that the authors demonstrate binding of TSLP to TSLP in the presence and absence of BP79.

Our Response: We fully agree with the reviewer's comment and would like to point out that we have indeed provided this pivotal data set in Fig. 4a and Fig. S4. In fact, we conducted a proximity ligation assay to demonstrate the inhibition of TSLP-TSLPR interactions in the presence and absence of BP79.

3. Use of Ex Vivo Systems:

Reviewer: The authors need to use ex vivo systems that more accurately reflect TSLP function for their analyses of BP79, etc. CD4 T cells generally do not express TSLPR until activated, and the role of TSLP in cytokine production is unclear. Keratinocytes are TSLP producers and not generally known as responders. The authors need to examine the ability of these molecules to inhibit antigen-presenting cell-mediated Th2 differentiation and TSLP-mediated CCL17 production.

Our Response: We appreciate the reviewer's suggestion and would like to clarify that our atopic diseases on-a-chip ex vivo system effectively mimics TSLP function. In AD, the downregulation of barrier proteins like filaggrin triggers the secretion of the inflammatory long-form TSLP. This inflammatory environment attracts activated CD4+ T cells from lymph nodes, and TSLP enhances Th2 cytokine secretion from these CD4+ T cells. This microenvironment also induces periostin expression from keratinocytes, contributing to itching and worsening AD.

Our organ-on-a-chip system faithfully replicates key atopic conditions. We knocked down filaggrin and induced TSLP expression in primary keratinocytes (Fig. 5b), creating an atopic dermatitis skin condition in a 3D tissue model. We activated primary human CD4+ T cells using CD3/CD28 antibodies, simulating dendritic cell-mediated activation, and introduced them into the microfluidic circulation circuit, mimicking blood flow.

While recapitulating the complexity of AD in a human-based system is indeed challenging, this is one of the best systems that are currently available as animal models almost entirely fail to emulate human AD. Also, studying the effect of TSLP and TSLP inhibitors in rodents is close to impossible given the lack of functional homology.

Additionally, the reviewer's comment about keratinocytes is not entirely accurate. Keratinocytes both produce TSLP and express TSLPR; this is well documented in the literature. Furthermore, the statement that "CD4 T cells generally do not express TSLPR until activated" is also inaccurate. TSLPR is constitutively expressed in CD4+ T cells, although its expression levels may increase upon activation. This distinction underscores the suitability of the TSLP signaling pathway as a target for drug intervention in Th2-mediated atopic diseases compared to the IL-33 signaling pathway, where ST2 expression is inducible only after activation.

Based on that, we would highly appreciate if you could reconsider the rejection of our manuscript without providing us with the chance to revise and resubmit. Overall, we feel that none of the comments raised are major concerns or actual flaws in our study. Reviewer 2 raises relevant questions, which, however, can be easily addressed in one

Page 4/4

round of revision as it's largely a re-analysis of existing data sets as well as some additional experiments.

I am looking forward to your reply. Please let me know in case further clarification may be required.

Regards,

Sarah

References

1. Adhikary, P.P., et al., *TSLP as druggable target - a silver-lining for atopic diseases?* Pharmacol Ther, 2021. **217**: p. 107648.
2. Dai, X., et al., *TSLP Impairs Epidermal Barrier Integrity by Stimulating the Formation of Nuclear IL-33/Phosphorylated STAT3 Complex in Human Keratinocytes.* J Invest Dermatol, 2022. **142**(8): p. 2100-2108.e5.
3. Demehri, S., et al., *Skin-derived TSLP triggers progression from epidermal-barrier defects to asthma.* PLoS Biol, 2009. **7**(5): p. e1000067.
4. Han, H., F. Roan, and S.F. Ziegler, *The atopic march: current insights into skin barrier dysfunction and epithelial cell-derived cytokines.* Immunol Rev, 2017. **278**(1): p. 116-130.
5. Han, H., et al., *Thymic stromal lymphopoietin (TSLP)-mediated dermal inflammation aggravates experimental asthma.* Mucosal Immunol, 2012. **5**(3): p. 342-51.
6. Ito, T., Y.J. Liu, and K. Arima, *Cellular and molecular mechanisms of TSLP function in human allergic disorders--TSLP programs the "Th2 code" in dendritic cells.* Allergol Int, 2012. **61**(1): p. 35-43.
7. Leyva-Castillo, J.M., et al., *TSLP produced by keratinocytes promotes allergen sensitization through skin and thereby triggers atopic march in mice.* J Invest Dermatol, 2013. **133**(1): p. 154-63.
8. Soumelis, V., et al., *Human epithelial cells trigger dendritic cell mediated allergic inflammation by producing TSLP.* Nat Immunol, 2002. **3**(7): p. 673-80.
9. Takahashi, N., et al., *Thymic Stromal Chemokine TSLP Acts through Th2 Cytokine Production to Induce Cutaneous T-cell Lymphoma.* Cancer Research, 2016. **76**(21): p. 6241-6252.
10. Wallmeyer, L., et al., *TSLP is a direct trigger for T cell migration in filaggrin-deficient skin equivalents.* Sci Rep, 2017. **7**(1): p. 774.

29th Sep 2023

Dear Dr. Hedtrich,

Thank you for your e-mail asking us to reconsider our decision on your manuscript and for providing a detailed provisional point-by-point letter, and please accept my apologies for the delay in getting back to you as I was waiting to get further advice on your manuscript. I have now received the feedback from two independent experts. As you will see below, they are overall supportive of your study, and after discussion with my colleagues, we would like to invite you to submit your revised work to EMBO Molecular Medicine.

Advisor 1 was asked to comment on your appeal letter, and particularly on the concerns from referee #1:

Advisor 1:

"The mechanistic aspect is not very strong, but publication may be justified because of the translational relevance.

Regarding the comments of reviewer 1:

Comments 1 and 3: I think they are partially justified. Keratinocytes are the major producers of TSLP and therefore, they of course have to be included in an assay. However, the reviewer is mainly concerned about the targets. Keratinocytes are also targets of TSLP action and CD4 cells as well. Therefore, the selection of the cells is fine. However, the major targets are dendritic cells, and the authors could include experiments with cultured dendritic cells to address the comment of the reviewer. The 3D system and the microfluidic system are good (although they may consider including dendritic cells).

Comment 2: In my opinion, it is indeed important to perform an in vitro receptor binding assay, e.g. with radiolabeled TSLP. Proximity ligation assay only shows that the different proteins are in close proximity. It is not even clear from Fig. 4 if the difference is statistically significant (probably not, because the significance is not indicated). N number is also missing (same problem in most other figures).

Other points:

A proper quantitative/statistical analysis of the data is important. For example, all Western blots should be done in triplicates and the data should be quantified. N numbers are frequently missing in the legend (see for example Fig. 4a, Fig. 5c, Fig. 6d and others).

The filaggrin staining in Fig. 5b is not convincing. It seems that there is only background staining in the cornified layer. Magnification bars are missing in this figure."

We also consulted a second advisor to comment on the drug development part:

Advisor 2:

"Hedtrich and coworkers report the development of a first-in-class TSLP-TSLPR interaction inhibitor and its broad application in in vitro models. The development of this compound is a prime example of pharmacophore- and docking-based drug discovery. Compound BP79 obtained after several cycles of in silico screening was comprehensively profiled in vitro and confirmed to exhibit cellular target engagement. It does not reach chemical probe quality but was used in a chemogenomics fashion as a set with matched negative control compounds. Selectivity screening has focused on kinases which might be considered a bit narrow, and the intermediate potency of the hit scaffold should be systematically optimized in future work. Overall, the ligand development for this orphan target is excellent and compound selection for biological studies is very well done. The manuscript is well written, experimental procedures are detailed and sound. I only have few comments (see below) and recommend publication after minor revision.

- Target inhibition data (e.g., Fig 1e,f; Fig 3a,b) should be fitted to determine IC₅₀, max. efficacy and slope. The lead compound BP79 should be accurately described for its potency based on the results. Currently, it is just denoted as "potent" which may be considered as a stretch given that its IC₅₀ is around 1-3 μM according to Fig 3a,b. The authors should also consider that the assays that was used to test the compounds is indirect and cellular. Affinity of BP79 to the target protein might be substantially higher.

- The dose-responses seem to have a quite steep slope. The authors should comment on this and confirm that no toxicity or aggregation occurs.

- Some of the tested compounds contain reactive or unstable motifs. The authors should comment on this and confirm that they have considered false positive and false negative assay results.

- The chemical structures are too small throughout the manuscript.

- Several references lack details on Journal, Publication year, etc."

In a revised manuscript, we would like you to address the initial referees' concerns (taking into account the comments from Advisor #1), as well as the issues raised by both advisors. The revised manuscript will once again be subject to review.

EMBO Molecular Medicine encourages a single round of revision only and therefore, acceptance or rejection of the manuscript will depend on the completeness of your responses included in the next, final version of the manuscript. For this reason, and to save you from any frustrations in the end, I would strongly advise against returning an incomplete revision.

We are expecting your revised manuscript within three months, if you anticipate any delay, please contact us.

We require:

4) A .docx formatted letter INCLUDING the reviewers' reports and your detailed point-by-point responses to their comments. As part of the EMBO Press transparent editorial process, the point-by-point response is part of the Review Process File (RPF), which will be published alongside your paper.

5) A complete author checklist, which you can download from our author guidelines (<https://www.embopress.org/page/journal/17574684/authorguide#submissionofrevisions>). Please insert information in the checklist that is also reflected in the manuscript. The completed author checklist will also be part of the RPF.

6) Please note that all corresponding authors are required to supply an ORCID ID for their name upon submission of a revised manuscript. An ORCID identified is currently missing for Brent Page.

7) It is mandatory to include a 'Data Availability' section after the Materials and Methods. Before submitting your revision, primary datasets produced in this study need to be deposited in an appropriate public database, and the accession numbers and database listed under 'Data Availability'. Please remember to provide a reviewer password if the datasets are not yet public (see <https://www.embopress.org/page/journal/17574684/authorguide#dataavailability>).

In case you have no data that requires deposition in a public database, please state so in this section ("This study includes no data deposited in external repositories."). Note that the Data Availability Section is restricted to new primary data that are part of this study.

8) For data quantification: please specify the name of the statistical test used to generate error bars and P values, the number (n) of independent experiments (specify technical or biological replicates) underlying each data point and the test used to calculate p-values in each figure legend. The figure legends should contain a basic description of n, P and the test applied. Graphs must include a description of the bars and the error bars (s.d., s.e.m.). Please provide exact p values.

- Additional Tables/Datasets should be labeled and referred to as Table EV1, Dataset EV1, etc. Legends have to be provided in

a separate tab in case of .xls files. Alternatively, the legend can be supplied as a separate text file (README) and zipped together with the Table/Dataset file.

13) Author contributions: CRediT has replaced the traditional author contributions section because it offers a systematic machine readable author contributions format that allows for more effective research assessment. Please remove the Authors Contributions from the manuscript and use the free text boxes beneath each contributing author's name in our system to add specific details on the author's contribution. More information is available in our guide to authors.

16) As part of the EMBO Publications transparent editorial process initiative (see our Editorial at <http://embomolmed.embopress.org/content/2/9/329>), EMBO Molecular Medicine will publish online a Review Process File (RPF) to accompany accepted manuscripts.

In the event of acceptance, this file will be published in conjunction with your paper and will include the anonymous referee reports, your point-by-point response and all pertinent correspondence relating to the manuscript. Let us know whether you agree with the publication of the RPF and as here, if you want to remove or not any figures from it prior to publication. Please note that the Authors checklist will be published at the end of the RPF.

I look forward to receiving your revised manuscript.

Yours sincerely,

Lise Roth

Point-by-point answers

We would like to thank the reviewers, advisors, and editors for their thorough assessment and the constructive criticism of our manuscript which helped further strengthening our manuscript. We now provide a revised version that was prepared according to these valuable suggestions and included several new data sets which further corroborate our claims. The modifications in the main text and supplemental material are highlighted in yellow. Please see our point-by-point responses below.

Advisor #1

Comments 1 and 3: I think they are partially justified. Keratinocytes are the major producers of TSLP and therefore, they of course have to be included in an assay. However, the reviewer is mainly concerned about the targets. Keratinocytes are also targets of TSLP action and CD4 cells as well. Therefore, the selection of the cells is fine. However, the major targets are dendritic cells, and the authors could include experiments with cultured dendritic cells to address the comment of the reviewer. The 3D system and the microfluidic system are good (although they may consider including dendritic cells).

Reply: Thank you for your assessment, which we fully agree with. Based on this, we have now included a new data set data that demonstrates efficient suppression of DC-relevant chemokines such as CCL17 after BP79 treatment. We further observed distinctly reduced OX-40L expression and consequentially, TSLP-mediated IL-13 release from co-cultivated T cells (Fig. 3I-K). We discuss this further in the manuscript in line 249-253.

As for the 3D systems: Fully agree that the inclusion of DC would be extremely valuable. However, the entire bioengineering field still struggles to stably incorporate functional human DCs. This still requires a lot of optimization and no major breakthrough has been achieved yet in this context.

Comment 2: In my opinion, it is indeed important to perform an in vitro receptor binding assay, e.g. with radiolabeled TSLP. Proximity ligation assay only shows that the different proteins are in close proximity. It is not even clear from Fig. 4 if the difference is statistically significant (probably not, because the significance is not indicated). N number is also missing (same problem in most other figures).

Reply: Thank you for this comment. We investigated several additional ways of doing this, but note there are no established methods to look at direct binding between TSLPR and experimental small molecules. Further, detecting/quantifying interactions between small molecules and proteins is generally very challenging. We refer the reviewer to Fig 4B, where our thermal shift assay results show a strong stabilizing effect of BP79 on recombinant TSLPR. This demonstrates direct binding to TSLPR.

Nevertheless, we conducted additional SPR measurements using a Biacore instrument for the revision, which however did not produce conclusive results. This may be due to the following reasons: (1) TSLPR and IL7R are transmembrane receptors. While recombinant proteins are commercially available, the quality, purity, and functionality of these proteins is highly variable. It is possible that they do not fold correctly compared to the native state, thus are not suitable to assess the binding of our small molecules. (2) These recombinant proteins are stabilized through either a His-tag or Fc-tag, which however may also impact the protein structure, thus not resembling the native state.

Overall, we believe that the combination of in vitro thermal shift assay, proximity ligation assay and thorough analysis of additional chemical inhibitors provide strong supporting evidence that their activity is due to disrupting TSLP-TSLPR interactions and subsequently

blocking downstream signaling events. However, to address the advisor's and reviewer's comments, we have added this explanation into the discussion and rephrased the manuscript, including its title, by referring to our molecules as 'putative small molecule-TSLP inhibitors' (line 409-418). We hope you will find this solution acceptable.

Other points: A proper quantitative/statistical analysis of the data is important. For example, all Western blots should be done in triplicates and the data should be quantified. N numbers are frequently missing in the legend (see for example Fig. 4a, Fig. 5c, Fig. 6d and others). The filaggrin staining in Fig. 5b is not convincing. It seems that there is only background staining in the cornified layer. Magnification bars are missing in this figure."

Reply: Thank you for raising this important point. We have now added the missing number of replicates where appropriate: Please note that all our experiments have been at least done in triplicates. We now also provide the requested semi-quantification of the WB data in Fig. 3G and H.

Advisor 2: Hedtrich and coworkers report the development of a first-in-class TSLP-TSLPR interaction inhibitor and its broad application in in vitro models. The development of this compound is a prime example of pharmacophore- and docking-based drug discovery. Compound BP79 obtained after several cycles of in silico screening was comprehensively profiled in vitro and confirmed to exhibit cellular target engagement. It does not reach chemical probe quality but was used in a chemogenomics fashion as a set with matched negative control compounds. Selectivity screening has focused on kinases which might be considered a bit narrow, and the intermediate potency of the hit scaffold should be systematically optimized in future work. Overall, the ligand development for this orphan target is excellent and compound selection for biological studies is very well done. The manuscript is well written, experimental procedures are detailed and sound. I only have few comments (see below) and recommend publication after minor revision.

Reply: We would like to thank you for this overall positive assessment.

Comment 1: Target inhibition data (e.g., Fig 1E-F; Fig 3A-B) should be fitted to determine IC₅₀, max. efficacy and slope. The lead compound BP79 should be accurately described for its potency based on the results. Currently, it is just denoted as "potent" which may be considered as a stretch given that its IC₅₀ is around 1-3 μ M according to Fig 3A-B. The authors should also consider that the assays that was used to test the compounds is indirect and cellular. Affinity of BP79 to the target protein might be substantially higher.

Reply: Thank you for this comment. We have now added the IC₅₀ value and slope in Figure 1 and 3 and the fitted curves in the Appendix, Fig. S2B for C7 and C13 and EV2 for BP79.

Comment 2: The dose-responses seem to have a quite steep slope. The authors should comment on this and confirm that no toxicity or aggregation occurs.

Reply: Thank you for bringing this point up, we highlight the cell viability data for these compounds compared to their inhibition of IL4 and IL13 expression. While we agree that the inhibition of cytokine expression curves appear to be steep end in their bar graph format, we believe that this is in part an artifact of how the data is displayed, as in an xy plot of concentration versus cytokine level, these higher concentration points would be much more spaced out resulting in less steep dose responses. Fig 1G-I shows minimal impact on viability for C7, 13 and 14, in Hut78 cells, primary keratinocytes and primary fibroblasts, whereas Fig 3 C-E shows minimal impact on viability of CD4+ T-cells, primary keratinocytes and primary fibroblasts for BP75 and BP79.

Comment 3: Some of the tested compounds contain reactive or unstable motifs. The authors should comment on this and confirm that they have considered false positive and false negative assay results.

Reply: This is a great point. Indeed, BP79 contains alpha-beta unsaturated carbonyl groups that would be considered vulnerable to nucleophilic attack as Michael acceptors. Indeed, we had considered this with respect to the mechanism of action for the compounds and considered that they may be acting as covalent modifiers of TSLPR. To address this point, we have included an additional series of BP79 analogues, all of which maintain the alpha-beta unsaturated carbonyl moiety of the parent compound (Fig. EV1 & EV2). However, we note that not all of these compounds were active in our TSLPR inhibition assays and varied in their ability to inhibit IL4 and 13 expression. We have expanded our commentary on this within the text, both acknowledging that this reactivity is possible, and also discussing the likelihood of these compounds acting via a covalent inhibitory mechanism. Please see p 8 (line 220-235) in our manuscript.

Comment 4: The chemical structures are too small throughout the manuscript.

Reply: We agree with that. However, it is quite challenging to put all this data together in a nice way and still to represent it in an adequate size. We have now tried to improve the legibility throughout the manuscript and hope you will find it acceptable.

Comment 5: Several references lack details on Journal, Publication year, etc."

Reply: Thanks for pointing this out, we have now revised that accordingly.

Referee #1 (Remarks for Author):

Comment 1. Based on the author's description, BP79 (the leading candidate) should act as a binding inhibitor of TSLP to TSLPR. It is important that the authors demonstrate binding of TSLP to TSLP in the presence and absence of BP79.

Reply: Thank you for this comment. We investigated several additional ways of doing this, but note there are no established methods to look at direct binding between TSLPR and experimental small molecules. Further, detecting/quantifying interactions between small molecules and proteins is generally very challenging. We refer the reviewer to Fig 4B, where our thermal shift assay results show a strong stabilizing effect of BP79 on recombinant TSLPR. This demonstrates direct binding to TSLPR.

Nevertheless, we conducted additional SPR measurements using a Biacore instrument for the revision, which however did not produce conclusive results. This may be due to the following reasons: (1) TSLPR and IL7R are transmembrane receptors. While recombinant proteins are commercially available, the quality, purity, and functionality of these proteins is highly variable. It is possible that they do not fold correctly compared to the native state, thus are not suitable to assess the binding of our small molecules. (2) These recombinant proteins are stabilized through either a His-tag or Fc-tag, which however may also impact the protein structure, thus not resembling the native state.

Overall, we believe that the combination of in vitro thermal shift assay, proximity ligation assay and thorough analysis of additional chemical inhibitors provide strong supporting evidence that their activity is due to disrupting TSLP-TSLPR interactions and subsequently blocking downstream signaling events. However, to address the advisor's and reviewer's comments, we have rephrased the manuscript, including its title by calling our molecules 'putative small molecule-TSLP inhibitors'. We hope you will find this solution acceptable.

Comment 2. The authors need to use ex vivo systems that more accurately reflect TSLP

function for their analyses of BP79, etc. CD4 T cells generally do not express TSLPR until activated, and the role of TSLP in cytokine production is unclear. Keratinocytes are TSLP producers, and not generally known as responders. The authors need to examine the ability of these molecules to inhibit antigen-presenting cell-mediated Th2 differentiation and TSLP-mediated CCL17 production.

Reply: We respectfully but strongly disagree with this statement. We deliberately selected keratinocytes and activated CD4+ T cells as primary cell types for specific reasons. The objective of our project was to develop a small molecule TSLP inhibitor for the treatment of atopic dermatitis (AD), a Th2-dominated condition in which TSLP is a central player.

In AD, skin barrier dysfunction occurs due to factors such as filaggrin gene mutations, microbial infections, or mechanical disruption. In this context, keratinocytes are the main source of TSLP, which triggers further inflammation. TSLP secreted by keratinocytes activates CD4+ T cells, either directly or through dendritic cells. These activated CD4+ T cells subsequently produce Th2 cytokines, including IL-4, IL-5, IL-9, and IL-13. Although other immune cells can also produce Th2 cytokines, CD4+ T cells are the primary source. Moreover, TSLP plays a crucial role in differentiating CD4+ T cells into pathogenic Th2 cells. In summary, keratinocytes produce TSLP and activate CD4+ T cells, which, in turn, produce Th2 cytokines. There is a large body of evidence corroborating these claims [1-10] and, thus, fully justifying the use of these cell types for drug discovery in AD. Further, we cannot think of cell types that may be better suited for our purpose. The following publications corroborate our statement:

References:

1. Adhikary, P.P., et al., *TSLP as druggable target - a silver-lining for atopic diseases?* Pharmacol Ther, 2021. **217**: p. 107648.
2. Dai, X., et al., *TSLP Impairs Epidermal Barrier Integrity by Stimulating the Formation of Nuclear IL-33/Phosphorylated STAT3 Complex in Human Keratinocytes.* J Invest Dermatol, 2022. **142**(8): p. 2100-2108.e5.
3. Demehri, S., et al., *Skin-derived TSLP triggers progression from epidermal-barrier defects to asthma.* PLoS Biol, 2009. **7**(5): p. e1000067.
4. Han, H., F. Roan, and S.F. Ziegler, *The atopic march: current insights into skin barrier dysfunction and epithelial cell-derived cytokines.* Immunol Rev, 2017. **278**(1): p. 116-130.
5. Han, H., et al., *Thymic stromal lymphopoietin (TSLP)-mediated dermal inflammation aggravates experimental asthma.* Mucosal Immunol, 2012. **5**(3): p. 342-51.
6. Ito, T., Y.J. Liu, and K. Arima, *Cellular and molecular mechanisms of TSLP function in human allergic disorders--TSLP programs the "Th2 code" in dendritic cells.* Allergol Int, 2012. **61**(1): p. 35-43.
7. Leyva-Castillo, J.M., et al., *TSLP produced by keratinocytes promotes allergen sensitization through skin and thereby triggers atopic march in mice.* J Invest Dermatol, 2013. **133**(1): p. 154-63.
8. Soumelis, V., et al., *Human epithelial cells trigger dendritic cell mediated allergic inflammation by producing TSLP.* Nat Immunol, 2002. **3**(7): p. 673-80.
9. Takahashi, N., et al., *Thymic Stromal Chemokine TSLP Acts through Th2 Cytokine Production to Induce Cutaneous T-cell Lymphoma.* Cancer Research, 2016. **76**(21): p. 6241-6252.
10. Wallmeyer, L., et al., *TSLP is a direct trigger for T cell migration in filaggrin-deficient skin equivalents.* Sci Rep, 2017. **7**(1): p. 774.

Nevertheless, we have now included additional data sets on human, primary DCs that demonstrate efficient suppression of DC-relevant chemokines such as CCL17 and the expression of OX-40L after BP79 treatment as well as IL-13 release from T cells following co-cultivation experiments with dendritic cells after BP79 treatment. This clearly verifies that BP79 also effectively suppresses TSLP-mediated effects in dendritic cells (Fig. 3I-K, line 249-253).

In addition, we would like to clarify that our atopic diseases on-a-chip ex vivo system effectively mimics TSLP function. In AD, the downregulation of barrier proteins like filaggrin

triggers the secretion of the inflammatory long-form TSLP. This inflammatory environment attracts activated CD4+ T cells from lymph nodes, and TSLP enhances Th2 cytokine secretion from these CD4+ T cells. This microenvironment also induces periostin expression from keratinocytes, contributing to itching and worsening AD.

Our organ-on-a-chip system faithfully replicates key atopic conditions. We knocked down filaggrin and induced TSLP expression in primary keratinocytes (Fig. 5B), creating an atopic-like skin condition in a 3D tissue model. We activated primary human CD4+ T cells using CD3/CD28 antibodies, simulating dendritic cell-mediated activation, and introduced them into the microfluidic circulation circuit, mimicking blood flow.

While recapitulating the complexity of AD in a human-based system is indeed challenging, this is one of the best systems that are currently available as most animal models fail to properly emulate human AD. Finally, studying the effect of TSLP and TSLP inhibitors in rodents is very challenging given the lack of functional homology.

Referee #2 (Remarks for Author):

Comment 1: Whereas it has been well shown that BP79 suppresses Th2 cytokines such as IL-13 and IL-4, phenotypical evidence is not sufficient. In Fig.6-c, the skin histopathology of the atopic condition treated with BP79 still seems to have hyperkeratosis and acanthosis. The authors need to show more convincing quantitative data on the phenotypic improvement by BP79.

Reply: Thank you very much for the comment. While we agree, we would like to highlight that this is very difficult to achieve with such 3D skin models as they do not undergo the same skin regeneration that occurs *in vivo*. Also, these 3D models do not desquamate. Hence, while we do see phenotypic differences, a full regeneration and the observation of improvement over time is not possible the same way it is in an *in vivo* setting. Nevertheless, we have now included an additional mRNA data set that shows the regulation of critical skin differentiation and proliferation markers following BP79 treatment which further corroborates its beneficial effects (Fig. 6E). We hope you will find this sufficient.

Comment 2: The authors claimed that BP79 suppressed immune cell skin infiltration in the abstract. However, they only showed immunohistochemistry of the human atopic-like skin disease model (AD-model) in Figure 6F, demonstrating that AD-model treated by BP79 has few CD4+ T cell skin infiltration, while the AD-model had rich CD4+ T cell skin infiltration. The authors need to show the quantitative data on these findings.

Reply: Thank you very much for this comment. As suggested, we have now included the requested quantitative data in Fig. 6H.

Comment 3: In addition, these results may be due to suppression of CD4+ T cell proliferation or induction of apoptosis of CD4+ T cells by BP79. The authors should explain how BP79 suppressed immune cell infiltration into the skin.

Reply: Thank you for this comment. In Fig. 3F, we show that BP79 indeed inhibits T cell proliferation. Another putative mechanism, however, is that TSLP directly stimulates T cell migration even in the absence of dendritic cells (Wallmeyer et al., 2017), which consequentially is reduced when TSLP-mediated effects are abrogated. We are now discussing that in line 428-431.

Comment 4. Please discuss reasons why the efficacy of tezepelumab in atopic dermatitis is inadequate and limited. In contrast, in this study the authors demonstrated BP79 suppressed significantly AD inflammation. This discrepancy may give readers a little concern about whether BP79 indeed improves atopic dermatitis in clinical settings.

Reply: Thank you for this comment. In fact, to the best of our knowledge, the reasons for this is unknown. However, we have now revised this passage to reflect that (line 384-386).

Comment 5. To prove that the transcriptional profiles of BP79-treated diseased skin are more similar to healthy skin than untreated diseased skin, the author compared the number of DEGs. However, the difference in transcriptional profiles should be interpreted not by the number of DEGs but by the contents of DEGs. For example, did the transcriptome of cytokines associated with atopic dermatitis change with or without treatment with BP79? Alternatively, clustering, such as PCA, might allow for qualitative evaluation if healthy skin and treated skin were included in the same cluster, separate from untreated skin.

Reply: Thank you for this comment. As suggested, we have amended the text and added further verification that treated skin diseases models are more similar to healthy skin than untreated. To strengthen this part, we have now added a discordance/concordance plot along with a heat map of the top 25 genes from each of the comparisons (healthy vs treated, healthy vs untreated, treated vs untreated) (Appendix Fig. S7-9). We hope you will find these revisions acceptable.

Comment 6. Relating to the comment above, the interpretation of RNA-seq results requires some caution. The results of RNA-seq demonstrated the enriched KEGG-Ribosome pathway as Fig.S8 showed many genes of ribosomal proteins. However, ribosomal RNA is not used for general RNA-seq analysis and a high quantity of rRNA could affect the quality of the sample or analysis. If rRNA is removed during the analysis phase, different pathways may be found.

Reply: The observed enrichments in gene sets related to protein translation cannot be explained with variability in the rRNA content, as the used gene sets, such as the transcriptional modules (Zyla et al. 2019) or the GO BP gene sets only contain protein coding genes and do not contain rRNA encoding genes. Nonetheless, we have removed all genes related to ribosomes from our data set to address the reviewer's concern. Below, we compare the results with and without ribosomal genes for three contrasts. Note that the results are basically identical (Fig. 1). Since it did not affect the results, we decided not to include that in the manuscript.

Figure 1: Comparison of the results with and without ribosomal genes for three contrasts. Colors show the degree of agreement between the two results.

Minor points:

1. Y axis of Figure 6D is written as "% of expression." It is unclear whether this was the

expression of mRNA or protein. If this is mRNA expression, the immunohistochemistry of indicated treatments in Figure 6D will add stronger evidence of the efficacy of BP79.

Reply: Thank you for this comment and sorry for the confusion. We have now revised the caption of Figure 6 to make it clear that this refers to protein expression.

2. Can the authors quantify the CD4+ T cell density in Figure 6F?

Reply: Yes, as suggested we have now included this quantitative data in Fig. 6H.

3. Line 170, is "IL-7R α " a mistake for "IL-7Ra"?

Reply: No – alpha is correct.

4. In the legend for Fig3, there are two explanation sentences for Fig.3-f, the former seems to be inappropriate.

Reply: Thank you for bringing this to our attention. We have removed the redundant sentence.

References:

Wallmeyer L, Dietert K, Sochorová M, Gruber AD, Kleuser B, Vávrová K, Hedtrich S.: TSLP is a direct trigger for T cell migration in filaggrin-deficient skin equivalents. *Sci Rep.* 2017 Apr 4;7(1):774.

Zyla, Joanna, Michal Marczyk, Teresa Domaszewska, Stefan HE Kaufmann, Joanna Polanska, and January Weiner 3rd. 2019. "Gene Set Enrichment for Reproducible Science: Comparison of CERNO and Eight Other Algorithms." *Bioinformatics* 35 (24): 5146–54.

25th Apr 2024

Dear Dr. Hedtrich,

Thank you for the submission of your revised study. Your manuscript was sent back to the two advisors who were consulted on your appeal. As you will see below, they are mostly satisfied with the revisions. I will therefore be able to accept your manuscript once the following points will be addressed:

1/ Please address the remaining concerns from referee #3 (previously advisor 1).

2/Manuscript text:

- Please remove the yellow highlights, and only keep in track changes mode any new modification. Your manuscript was cross-checked with previously published material, and similarities were found with your previous work (please see screenshot attached). Please kindly reword this paragraph.
- Materials and Methods:
 - o Cells: indicate whether the cells were authenticated and tested for mycoplasma contamination.
 - o Statistics: please include a statement on sample size, blinding, randomization, exclusion/inclusion criteria.
- Data availability: Please note that this section is meant to list large primary datasets produced in this study (sequencing, metabolomics, etc), and that need to be deposited in an appropriate public database. Source Data that are also provided with the manuscript should not be listed here. Please correct accordingly (<https://www.embopress.org/page/journal/17574684/authorguide#datadeposition>).
- Acknowledgements: the funding information provided in the manuscript and in the submission system should be identical (currently Sanofi Genzyme Canada, Canada Graduate Scholarship, Faculty of Pharmaceutical Sciences at UBC and the BC Lung Foundation are missing in the submission system).
- "Competing interests" should be renamed "Disclosure statement and competing interests": we request authors to consider both actual and perceived competing interests. Please review the policy <https://www.embopress.org/competing-interests> and update your competing interests if necessary.
- The references should be listed in alphabetical order, with 10 authors listed before et al. DOIs should be removed for published articles.
- Figure legends should be after References.
- Table 1 should be moved after the main figure legends and a callout should be added.

3/ Figures and Appendix:

- Please provide exact p values, not a range, in the figures or in their legends.
- Please remove "Source data are available online for this figure." from the figure legends.
- Appendix: please add the "Appendix Data and Methods" to the main manuscript file, Materials and Methods. The Appendix needs a table of content and page numbers. The nomenclature should be corrected to "Appendix Table S1", etc., and "Appendix Figure S1", etc. The heading "Citations" at the end of the appendix should be changed to "References". Yellow highlights should be removed from the final version.
- Please address the queries from our data editors in the figure legends:
 1. Please note that a separate 'Data Information' section is required in the legends of figures 1a-i; 2c-f; 3a-h, j-k; 6d-e, h; EV 2a-b.
 2. Please indicate the statistical test used for data analysis in the legends of figures 6d-e, h, j.
 3. Please note that in figures 4a; 6h; there is a mismatch between the annotated p values in the figure legend and the annotated p values in the figure file that should be corrected.
 4. Please note that information related to n is missing in the legends of figures 3f; 6d; EV 1c; EV 2c.
 5. Although 'n' is provided, please describe the nature of entity for 'n' in the legends of figures 1c-i; 2c-f; 3a-e, g-h, j-k; 4a; 5c; 6e, h.
 6. Please note that the error bars are not defined in the legends of figures 6d-e; EV 1c; EV 2c.
 7. Please note that the scale bar needs to be defined for figures 6c, f-g.
 8. Please note that scale bar and its definition are missing for figure 4a; EV 3.

4/ Checklist:

- please fill in the section Cell materials/ mycoplasma and authentication
- please complete the section on 'Experimental study design and statistics'

5/ Please provide 'The paper explained': EMBO Molecular Medicine articles are accompanied by a summary of the articles to emphasize the major findings in the paper and their medical implications for the non-specialist reader. Please provide a draft summary of your article highlighting

- the medical issue you are addressing,
- the results obtained and

- their clinical impact.

6/ Synopsis: thank you for providing a nice synopsis picture. It should be resized to a png/jpeg/tiff file 550px wide x 300-600 pixels high. The text should remain legible.

Please also provide a synopsis text: Synopses are displayed on the journal webpage, they include a short stand first (maximum of 300 characters, including space) as well as 2-5 one-sentences bullet points that summarizes the paper (maximum of 30 words / bullet point).

7/ As part of the EMBO Publications transparent editorial process initiative (see our Editorial at <http://embomolmed.embopress.org/content/2/9/329>), EMBO Molecular Medicine will publish online a Review Process File (RPF) to accompany accepted manuscripts.

This file will be published in conjunction with your paper and will include the anonymous referee reports, your point-by-point response and all pertinent correspondence relating to the manuscript. Let us know whether you agree with the publication of the RPF and as here, if you want to remove or not any figures from it prior to publication.

I look forward to receiving your revised manuscript.

Yours sincerely,

Lise Roth

***** Reviewer's comments *****

Referee #3 (Remarks for Author):

The authors have addressed the criticisms of the reviewers and the advisors in this revised version, and the manuscript is further improved. It is a pity that the direct receptor binding could not be shown, but the indirect evidence for receptor binding is acceptable. The DC experiments are very helpful.

The third paragraph on page 11 should be modified:

a.) The authors did not show that the barrier is improved. Increased filaggrin and decreased TSLP mRNA and protein expression only suggest an effect on the barrier, but this has not been functionally tested - please modify.

b.) KRT5 and KRF14 are not proliferation markers - they are markers for non-differentiated, proliferation-competent cells. However, many KRT5/KRT14 positive cells do not proliferate - please modify.

Some of the figures are extremely small and should be enlarged for publication, e.g. the immunostainings. Some text is not readable.

I find the discussion very long - it includes a lot of repetition. I do not insist on a revision of the discussion, because this was not be requested in the first round of reviews, but the authors may consider it.

Referee #4 (Comments on Novelty/Model System for Author):

The authors have convincingly responded to my comments and resolved the issues.

Dear reviewer and editors,

We would like to thank you for the positive feedback regarding our revision. We have addressed the editorial comments as well as the points raised by referee #3. The changes are marked in yellow throughout the text. As for the exact p values, we decided to add these in table format to the appendix (now Appendix Table 10), otherwise we were worried that adding these to the figures would again impact the legibility.

We hope you will find these edits now satisfactory.

Referee 3:

The third paragraph on page 11 should be modified:

1. a.) The authors did not show that the barrier is improved. Increased filaggrin and decreased TSLP mRNA and protein expression only suggest an effect on the barrier, but this has not been functionally tested - please modify.

b.) KRT5 and KRF14 are not proliferation markers - they are markers for non-differentiated, proliferation-competent cells. However, many KRT5/KRT14 positive cells do not proliferate - please modify.

Reply: Thank you for these comments. We have revised this paragraph as follows:

“BP79 treatment also exerts beneficial effects on the epithelial barriers, as exemplified by increased filaggrin and decreased TSLP mRNA and protein expression in the skin and lungs (Fig. 6E, F, Appendix Fig. S5). On mRNA level, a significant decrease of KRT5 and KRT14, markers for non-differentiated, proliferation-competent cells, was also noted following BP79 and tacrolimus treatment...”

2. Some of the figures are extremely small and should be enlarged for publication, e.g. the immunostainings. Some text is not readable.

Reply: We have now decided to split Fig. 1 into two figures and did the same for Fig. 6. This helped us to enlarge the data sets in question and hope you will find this acceptable.

3. I find the discussion very long - it includes a lot of repetition. I do not insist on a revision of the discussion, because this was not be requested in the first round of reviews, but the authors may consider it.

Reply: Thank you for this comment. We have now shortened the discussion and removed redundant parts.

17th May 2024

Dear Dr. Hedtrich,

Thank you for submitting your revised files. Almost everything is fine now, but there are still a few editorial points to address before final acceptance:

- Please remove the yellow highlights, and only keep in track changes mode any new modification.
- Statistics: please note that information on randomization, inclusion/exclusion criteria and blinding must be included in the manuscript and in the checklist, even if no blinding/randomization were performed.
- Data availability: Please note that the datasets must be public before manuscript acceptance.
- Please note that scale bar and its definition are missing for figure 5a; EV 3.
- The paper explained: Let me know if you agree with the following edits, or amend as you see fit, and include in the manuscript text file:

Problem:

Thymic stromal lymphopoietin (TSLP) is a pro-inflammatory cytokine that drives many inflammatory diseases including eczema, which has sparked great interest for therapeutic targeting. However, so far very few drugs are available and all of them are proteins which are expensive and must be systemically administered.

Results:

We describe the development of the first putative, non-proteinic TSLP inhibitors which can also be applied locally onto the skin, thus significantly expanding the treatment options for eczema patients. We used a combined approach of structure-based virtual screening and docking of >1,000,000 compounds, chemical optimization and iterative biological testing focused on translational methodology. This led to the identification of our lead compound BP79 which effectively abrogates TSLP-mediated effects and can significantly down-regulate eczema-relevant proinflammatory pathways and cytokines.

Impact:

We report the first potent small molecule TSLPR inhibitor which has the potential to expand the therapeutic options in eczema and potentially even beyond.

- Synopsis: Let me know if you agree with the following edits or amend as you see fit:

A potent small molecule TSLPR inhibitor was identified as a novel therapeutic option for atopic diseases.

- The small molecule BP79 effectively abrogates TSLP-triggered immune reactions at low micromolar concentrations in human skin and lungs.
- BP79 can be topically applied and, thus, provides significant advantages over the current TSLP antibodies.
- A human-based atopic disease drug discovery platform was developed with the potential to facilitate preclinical development of atopy-targeting drugs.

I look forward to receiving your revised manuscript at your earliest convenience.

With kind regards,

Lise Roth

The authors addressed the minor editorial issues.

Dear reviewer and editors,

We would like to thank you for the positive feedback regarding our revision. We have addressed the editorial comments as well as the points raised by referee #3. The changes are marked in yellow throughout the text. As for the exact p values, we decided to add these in table format to the appendix (now Appendix Table 10), otherwise we were worried that adding these to the figures would again impact the legibility.

We hope you will find these edits now satisfactory.

Referee 3:

The third paragraph on page 11 should be modified:

1. a.) The authors did not show that the barrier is improved. Increased filaggrin and decreased TSLP mRNA and protein expression only suggest an effect on the barrier, but this has not been functionally tested - please modify.

b.) KRT5 and KRF14 are not proliferation markers - they are markers for non-differentiated, proliferation-competent cells. However, many KRT5/KRT14 positive cells do not proliferate - please modify.

Reply: Thank you for these comments. We have revised this paragraph as follows:

“BP79 treatment also exerts beneficial effects on the epithelial barriers, as exemplified by increased filaggrin and decreased TSLP mRNA and protein expression in the skin and lungs (Fig. 6E, F, Appendix Fig. S5). On mRNA level, a significant decrease of KRT5 and KRT14, markers for non-differentiated, proliferation-competent cells, was also noted following BP79 and tacrolimus treatment...”

2. Some of the figures are extremely small and should be enlarged for publication, e.g. the immunostainings. Some text is not readable.

Reply: We have now decided to split Fig. 1 into two figures and did the same for Fig. 6. This helped us to enlarge the data sets in question and hope you will find this acceptable.

3. I find the discussion very long - it includes a lot of repetition. I do not insist on a revision of the discussion, because this was not be requested in the first round of reviews, but the authors may consider it.

Reply: Thank you for this comment. We have now shortened the discussion and removed redundant parts.

23rd May 2024

Dear Dr. Hedtrich,

Thank you for submitting your revised files. I am pleased to inform you that your manuscript is accepted for publication and is now being sent to our publisher to be included in the next available issue of EMBO Molecular Medicine!

Please note that I modified your Checklist (Blinding and randomization/ is the information in the manuscript: yes; Materials and Methods) to reflect the information in your manuscript. Please let us know immediately if this is not correct.

If you have any questions, please do not hesitate to contact the Editorial Office.
Thank you for your contribution to EMBO Molecular Medicine!

With kind regards,

Lise
